# Oncolytic Tanapoxvirus Variants Expressing mIL-2 and mCCL-2 Regress Human Pancreatic Cancer Xenografts in Nude Mice [note 1]

**DOI:** 10.3390/biomedicines12081834

**Published:** 2024-08-12

**Authors:** Scott D. Haller, Karim Essani

**Affiliations:** Laboratory of Virology, Department of Biological Sciences, Western Michigan University, Kalamazoo, MI 49008-5410, USA; scott.haller@crl.com

**Keywords:** tanapoxvirus, pancreatic cancer, oncolytic virus, immuno-oncolytic virus

## Abstract

Pancreatic ductal adenocarcinoma (PDAC) is the fifth leading cause of cancer-related death and presents the lowest 5-year survival rate of any form of cancer in the US. Only 20% of PDAC patients are suitable for surgical resection and adjuvant chemotherapy, which remains the only curative treatment. Chemotherapeutic and gene therapy treatments are associated with adverse effects and lack specificity/efficacy. In this study, we assess the oncolytic potential of immuno-oncolytic tanapoxvirus (TPV) recombinants expressing mouse monocyte chemoattractant protein (mMCP-1 or mCCL2) and mouse interleukin (mIL)-2 in human pancreatic BxPc-3 cells using immunocompromised and CD-3^+^ T-cell-reconstituted mice. Intratumoral treatment with TPV/∆66R/mCCL2 and TPV/∆66R/mIL-2 resulted in a regression in BxPc-3 xenograft volume compared to control in immunocompromised mice; mCCL-2 expressing TPV OV resulted in a significant difference from control at *p* < 0.05. Histological analysis of immunocompromised mice treated with TPV/∆66R/mCCL2 or TPV/∆66R/mIL-2 demonstrated multiple biomarkers indicative of increased severity of chronic, active inflammation compared to controls. In conclusion, TPV recombinants expressing mCCL2 and mIL-2 demonstrated a therapeutic effect via regression in BxPc-3 tumor xenografts. Considering the enhanced oncolytic potency of TPV recombinants demonstrated against PDAC in this study, further investigation as an alternative or combination treatment option for human PDAC may be warranted.

## 1. Introduction

In this paper, we demonstrate regression in pancreatic cancer xenografts following tanapoxvirus variant(s) treatment in an immunocompromised murine model. We also demonstrate a novel nuclear imaging technique and analysis methodology. The significance of these key findings is (1) the potential application of immuno-oncolytic tanapoxvirus as a clinical treatment option and (2) novel application of PET imaging to assess proliferation activity within pancreatic cancer tumors clinically with improved accuracy and consistency of patient diagnosis, stagging, and monitoring. 

Given the severity, increasing prevalence, and extremely low survival rate associated with pancreatic cancer, as well as the lack of clinical treatment options and the challenges associated with non-invasive pancreatic cancer diagnosis and monitoring, we feel the results presented in our paper will appeal to a wide variety of scientific and healthcare professionals. 

Pancreatic cancer has one of the poorest clinical prognoses, with under a 10% five-year survival rate [1]. Pancreatic ductal adenocarcinoma (PDAC) is the most common form of pancreatic cancer, accounting for about 90% of all cases of pancreatic cancer [2,3,4]. By 2030, PDAC is expected to become the second leading cause of cancer-related deaths in the United States [5,6]. The primary reason for the low survival rate is a lack of direct or indirect diagnostic biomarkers for the disease, which leads to late-stage diagnosis, often preventing curative surgical resection [2,3]. Since complete resection is currently the only potential cure for PDAC, early detection is critical in the pursuit of increasing the median survival length of PDAC patients. 

PDACs evolve through genetic and lifestyle-related factors [7,8,9,10,11,12]. The most frequent genetic abnormalities of PDAC are mutational activation of the *KRAS* oncogene; inactivation of tumor-suppressor genes *CDKN2A*, *TP53*, *SMAD4*, and *BRCA2*; widespread chromosomal losses, gene amplifications, and telomere shortening [13,14,15,16,17,18]. Epigenetic modifications have also been associated with PDAC [19,20,21,22,23,24,25,26,27,28,29]. 

Staging of PDAC is facilitated via diagnostic imaging using computed tomography (CT), magnetic resonance imaging (MRI), endoscopic ultrasonography, and/or [^18^F]-fluorodeoxyglucose (FDG) positron emission tomography/computed tomography (PET/CT). Due to the complexity of the PDC tumor microenvironment, up to 20% of patients are staged incorrectly upon primary diagnosis [30,31]. 

Contemporaneous treatment is surgical resection with adjuvant chemotherapy, though this is limited to only ~15% of patients. Surgical resection has demonstrated marked improvement in patient survival in combination therapy with gemcitabine and capecitabine [31,32,33,34]. The majority of PDAC cases, 80–90%, comprise patients presenting with locally advanced, non-resectable tumors and systemically disseminated metastases [35,36,37,38]. 

Several additional characteristics of PDAC present challenges for most therapeutic approaches, such as the composition of the tumor microenvironment presenting as dense fibrotic stroma and stellate cells that prevent or inhibit access of intended therapeutic agents to proliferating cells and expression of immunosuppressive factors [39]. PDAC tumors also do not appear to express neoantigens; thus, immune system response to the tumor is limited [7]. As a result of these characteristics, treatment strategies involving the use of OVs present unique challenges for therapeutic efficacy. To overcome the specific challenges associated with the microenvironment, Vitamin D and hyaluronidase, in conjunction with therapeutic agents, are currently being investigated for the potential to increase direct exposure and facilitate enhanced therapeutic efficacy [40,41]. 

The term “oncolytic and immuno-oncolytic virus” (OV) genesis involves the discovery and potential use of differing naturally occurring or genetically modified viruses as therapeutic agents in the treatment of various forms of cancer. Most Ovs are non-pathogenic viral strains that demonstrate differing modes of selective replication in cancer cells over noncancerous cells [42,43,44,45]. As standard practice for the development of novel therapeutics, OVs have been assessed and have demonstrated efficacy in a regression in differing forms of cancer in preclinical models [46]. The mechanism of action (MOA) of OVs differs widely, such as direct malignant cell lysis, expression of cytotoxic or immunomodulatory genes, and inherent susceptibility of differing forms of cancer to viral replication [43,46,47]. The approach of genetically modifying wild-type (wt) virus to express immunomodulatory genes resulting in stimulation or suppression of the patient’s immune system results in an immunogenically “hot” environment around the tumor, which promotes regression in the malignant cell population [40,48,49].

There is a growing number of ongoing pre-clinical and clinical research programs assessing the potential of various OV platforms as a potential treatment of PDAC. Such programs comprise various families of viruses, including but not limited to adenovirus (AV), herpes simplex virus (HSV), parvovirus (PV), reovirus (RV), vaccinia virus (VV), and protoparvovirus (PV) [50,51,52,53,54,55,56,57,58,59,60,61,62,63,64,65,66,67].

Though the OV platforms of AV, HSV, PV, and RV provide promising potential, the family of poxviruses presents an inherent advantage given cancerous cells demonstrate greater vulnerability to poxviruses due to ineffective anti-viral, innate immune response pathways, unlike non-cancerous cells, which respond acutely to poxvirus infection and impede viral replication [68]. Tanapoxvirus (TPV) is a DNA virus classified in the family *Poxviridae* and is a member of the genus *Yatapoxvirus*. TPV is a potentially suitable OV candidate, as pre-existing exposure and immunity in the general human population is limited since TPV exposure is restricted to equatorial Africa, has lack of cross-reactivity with other poxviruses, and causes mild illness with limited human-to-human transmission [69,70,71]. Furthermore, the large genome of TPV comprises approximately 145 kb, providing numerous transgenic modification sites to result in cell selectivity and immune modulation intended to enhance the oncolytic activity of TPV [72,73,74,75,76,77,78,79,80].

Several TPVs have been engineered with specific genetic modifications of wt TPV to generate recombinants with the intent to investigate this virus as an OV with enhanced research applications and oncolethality. Such genetic modifications include TPV *thymidine kinase* (TK) *66R*, *2L* (a high-affinity inhibitor of human tumor necrosis factor (TNF)), and *TPV-15L* (a functional mimic of the neuregulin (NRG) that acts through the ErbB family of tyrosine kinase receptors) gene knockouts and monocyte chemoattractant protein (mMCP-1; also known as mCCL2), mouse interleukin (mIL)-2, enhanced green fluorescent protein (eGFP), and monomeric red fluorescent protein (mCherry) gene knock-ins [73,74,75]. A number of these TPV recombinants have also been assessed for preclinical therapeutic efficacy against various forms of cancer, including human colorectal cancer, melanoma, and triple negative breast cancer (TNBC) xenografts in nude mice [81,82,83].

Herein, we describe our assessment of the oncolytic potential of three TPV recombinants, TPV/eGFP, TPV/∆66R/m-CCL-2/mCherry, and TPV/∆66R/m-IL-2/mCherry, against human pancreatic cancer xenografts in an immunodeficient, athymic nude mouse model and an immunocompetent, athymic nude mouse model following adoptive transfer of CD3^+^ T cells. TPV/eGFP was used as a “wt” control virus and TPV/∆66R/m-CCL-2/mCherry and TPV/∆66R/m-IL-2/mCherry were selected specifically to assess mCCL-2 and mIL-2 immunomodulatory activity in combination with TPV oncolytic activity against pancreatic cancer due to the known biological activity of mCCL-2 and mIL-2 [84,85,86,87,88,89,90,91,92,93].

The athymic nude murine model has been the predominant preclinical model used by investigators in our laboratory [81,82,83,94]. The athymic mouse model is an immunodeficient preclinical model due to the lack of peripheral T lymphocytes and a functional adoptive immune system, although immature CD3^+^ T cells have been reported [83]. Yet the athymic nude mouse has an intact and functional innate immune system with active macrophages, NK cells, granulocytes, B cells, and dendritic cells. This model also has a compensatory increase in the population of anti-tumor macrophages and NK cells, with macrophages being the predominant mononuclear cell type [95,96]. Thus, the athymic nude mouse model was determined be to an appropriate and acceptable preclinical model to assess the oncolytic potential of TPV recombinants expressing mCCL-2 and mIL-2 in this study. In this study, we also assessed the oncolytic potential of the TPV recombinants in cohorts of immunocompromised athymic nude mice that received an adoptive transfer of CD3^+^ T cells following isolation and purification from immunocompetent BALB-c donor animals. The recipient animal strain was selected to be BALB/c (CanN.Cg-*Foxn1^nu^/Crl)* aythymic nude mice, as this strain is genetically identical to normal BALB/c mice at the immunological compatibility level. This approach allowed direct assessment and comparison of results of the TPV recombinants’ efficacy between mirrored cohorts of animals with either an immunocompromised or an immune-reconstituted immune system.

Our results demonstrate that TPV recombinants expressing mCCL-2 and mIL-2 were more effective in the regression in BxPc-2 pancreatic tumor volume and tumor proliferation rate compared to the TPV/eGFP control recombinant, with the mCCL-2 recombinant reaching statistical significance compared to vehicle-treated control animals in immunocompromised athymic nude mice. The results in all cohorts of immune-reconstituted mice receiving adoptive CD3+ T-cell transfer were indicative of immune system rejection of the BxPc-3 xenografts.

## 2. Methods and Materials

### 2.1. Cells, Viruses, and Reagents

Human pancreatic ductal adenocarcinoma BxPc-3 cells were purchased from American Type Culture Collection (ATCC, Rockville, MD, USA; ATCC CRL-1687^TM^). BxPc-3 cells were cultured in a traditional cell incubator maintained at 37 °C and supplemented with 5% CO_2_ using vented-top tissue culture flasks. Cells were maintained in growth media consisting of RPMI 1640 supplemented with 2 mM L-glutamine and 10% fetal bovine serum (FBS). Cells were allowed to reach approximately 85–90% confluence. When BxPc-3 cells reached 85–90% confluence, each flask was treated with dissociation reagent and held at 37 °C for approximately 5 min. Cells were split 1:3 in fresh tissue culture flasks. Sufficient cell population was maintained as needed to provide approximately 5 × 10^6^ cells/subject for xenograft inoculation. Wild-type (wt) TPV was originally received as a gift from Dr. Joseph Esposito (Centers for Disease Control, Atlanta, GA, USA) and was used to generate the TPV recombinants used in this study. Generation of the recombinants and confirmation of expression of each transgene by Western blot and/or ELISA analysis and *TK* gene expression confirmation by PCR has been previously described [81,83,94].

### 2.2. Viral Replication Assessment via Phase Contrast and Florescent Microscopy

BxPc-3 cells were cultured in growth media and allowed to reach 85–90% confluence. Once desired confluence was achieved, cells were infected with 1–2 µL of 100X TPV/eGFP, TPV/∆66R/m-CCL-2/mCherry, or TPV/∆66R/m-IL-2/mCherry and maintained for 5–7 days post infection in maintenance media to assess viral plaque formation via phase contrast microscopy and immunofluorescence microscopy. Maintenance media consisted of a lower total concentration of FBS (2%) with all other supplements being identical to growth media, as previously described. Phase contrast white-light microscopy images were acquired to determine viral plaque formation as confirmation of viral replication; uninfected BxPc-3 cells served as controls. BxPc-3 cells cultured in glass-bottom dishes planned for immunofluorescent imaging were infected with each applicable TPV variant as described above; uninfected cells served as control. Following visual confirmation of viral plaque formation via phase contrast white-light imaging at 5–7 days post infection, each tissue culture dish was washed with phosphate buffered saline (PBS), stained with 4′,6-diamindino-2-phenylindole (DAPI) stain, and PBS maintained over the cell monolayer to maintain cell integrity to facilitate immunofluorescent imaging. Assessment of viral reporter gene expression was performed via immunofluorescence microscopy using excitation lasers applicable for DAPI, eGFP, and mCherry. Immunofluorescent images were collected on a confocal microscope with a camera using NIS Elements software version 5.21.01., Melville, NY, U.S.A.

### 2.3. Viral Plaque Assay

TPV recombinants TPV/eGFP, TPV/∆66R/m-CCL-2/mCherry, and TPV/∆66R/m-IL-2/mCherry replication competency was assessed via BxPc-3 cells via viral plaque assay as previously described [97]. Briefly, BxPc-3 cells monolayers grown in 6-well plates were infected with 200.0 μL/well of serial ten-fold dilutions of each 100X TPV recombinant, with each dilution being plated in duplicate. Following an adsorption period of approximately 1 h, overlay medium was added to each well and the plates were then incubated under appropriate conditions for 5–7 days. Following the 5–7-day incubation, the overlay medium was carefully aspirated and the cell monolayers were stained with 1% crystal violet and fixed with formaldehyde. The plates were then rinsed with distilled water and allowed to dry. The plaques were counted to determine plaque-forming units (pfu; reported units pfu/mL) with the following formula: [pfu = ((number of plaques in well-A + plaques in well-B)/2) × dilution factor for each duplicate of wells], then [number pfu (200.0 μL)] × 5 = 1.0 mL pfu.

### 2.4. Animal Model

Female BALB-c nude mice (CAnN.Cg-*Foxn1^nu^/Crl*) and BALB-c mice, approximately 4 or 6 weeks old upon receipt and initiation of the 10–16-day acclimation period, were received from Charles River Laboratories, Raleigh, North Carolina. All study-specific activities were conducted under an approved protocol by the Charles River Laboratories (CRL) Institutional Animal Care and Use Committee (CRL IACUC Protocol No. 999-872). All animals were given a detailed clinical observation examination prior to selection for this study.

### 2.5. In Vivo Study Design Summary

Our study incorporated numerous endpoints to thoroughly assess oncolytic efficacy and potential mechanism(s) of action of TPV recombinants in response to BxPc-3 human PDAC tumor xenografts in immunocompromised and immune-reconstituted BALB-c (CAnN.Cg-*Foxn1^nu^/Crl*) nude mice. Appendix A provides a succinct summary of the study design employed and primary endpoints for each treatment group of animals assigned to the study; Appendix A displays the temporal relationship of key study activities employed throughout the course of the study or all study groups using tumor volume data for immunocompromised control subjects treated with applicable vehicle only.

#### BxPc-3 Human PDAC Xenografts and Virotherapy

BxPc-3 cells were cultured as previously described herein. For each cohort of animals, a sufficient number of BxPc-3 cells were maintained and prepared for subcutaneous (SC) tumor xenografts by inoculation in a final formulation of 50:50 sterile PBS and Matrigel. Inoculations were performed as unilateral inoculations in the right flank region of each animal. Cell concentration and viability were assessed pre- and post inoculation using trypan blue staining and analyzed via automated cell counter. Cell viability acceptance criteria were established as ≥85% and cell concentration criteria were 5 × 10^6^ cells/animal in a total volume of 100 µL. Within three days of inoculation, total volume analysis of all inoculation sites/tumors was initiated and continued throughout the course of the study, with digital caliper measurements being conducted every three days. Measurements were conducted in dimensions of length, width, and height to determine the total volume for each tumor and reported as cubic millimeters units (mm^3^). Once tumors reached approximately 200 mm^3^, animals were randomly assigned to study treatment groups based on tumor volume to achieve equivalent mean group tumor volume as much as possible. Following animal assignment to control groups, animals were treated with a single intratumoral administration of 100 µL PBS formulation vehicle. Animals assigned to virotherapy groups were treated with a single intratumoral administration of 100X virus stock of each applicable recombinant virus diluted to a final volume of 100 µL with PBS formulation vehicle to deliver a viral mass dose of 5 × 10^6^ pfu per animal.

### 2.6. CD3^+^ T-Cell Isolation and Adoptive Transfer

Cohorts of immunocompromised BALB-c nude mice received adoptive transfer of isolated cluster of differentiation 3 (CD-3)-positive T-cells from donor immunocompetent BALB-c mice to assess oncolytic activity of TPV recombinants employed in this study in an immune-reconstituted murine model. CD-3^+^ T-cells were isolated and injected into the tail vein of all animals assigned to immune-reconstituted cohorts 4 days post virotherapy per the following method. Briefly, intact spleens from BALB-c donor mice were collected and processed to separate splenocyte phenotypes, and CD-3^+^ T-cells were isolated by CD-3 affinity column isolation. Isolation of CD-3^+^ T-cells was confirmed by FLOW cytometry. Immune-reconstituted study animals received adoptive T-cell transfer at a ratio of approximately 1:3 donors:recipients, respectively. Cell concentration was adjusted for the intended cell number of 5 × 10^6^ cells/subject via IV injection in a total volume of 200 μL/subject.

### 2.7. [^18^F]-FDG and [^18^F]-FLT PET Imaging Agents

[^18^F]-fluorodeoxyglucose ([^18^F]-FDG) was procured through a commercial vendor and used, neat, as received from the vendor to assess metabolic activity at the site of each tumor in each subject.

A novel radiolabeling method was established for production of [^18^F]-fluoro-3′-deoxy-3-L: -fluorothymidine ([^18^F]-FLT) based on an adaptation of method as previously described [98]. [^18^F]-FLT was produced via an automated synthesis module as needed to support PET imaging to assess tumor-site proliferation.

### 2.8. [^125^I]-Anti GFP Antibody and [^125^I]-Anti-mCherry Antibody SPECT Imaging Agents

Ig-G antibodies targeting green fluorescent protein (GFP) and monomeric red fluorescent protein (mCherry) were procured through a commercial vendor to enable in vivo SPECT/CT imaging of oncolytic virus reporter gene expression following [^125^I]-sodium iodine ([^125^I]-NaI) radio iodination radiolabeling of each applicable antibody to generate [^125^I]-anti-GFP and [^125^I]-anti-mCherry antibodies.

Following establishment of an acceptable radiolabeling method to produce [^125^I]-anti-GFP and [^125^I]-anti-mCherry antibodies of sufficient yield and radiochemical purity to enable SPECT/CT imaging, the stability of each labeled antibody was assessed and confirmed to be acceptable at 1, 24, and 48 h post radiolabeling.

The post-radiolabeling bioactivity of each antibody was also assessed by the ELISA method to confirm antigenic affinity for GFP, and mCherry was not reduced/negated through radio iodination radiolabeling of the antibodies. To enable the ELISA assay, nonradioactive forms of each antibody were produced through substitution of nonradioactive isotope ^127^I for radioactive isotope ^125^I and applying the same radio iodination method to produce radioactive ^125^I forms of anti-GFP and anti-mCherry antibodies. The ELISA method was performed as follows: Briefly, 96-well plates were coated with 50 µL/well of 4 µg/mL recombinant GFP or 4 µg/mL recombinant mCherry proteins. Plates were applicably washed and then blocked with 150 µL per well of 3% BSA in 1X PBS. [^127^I]-anti-GFP and non-modified GFP, [^127^I]-anti-mCherry and non-modified mCherry, and Mouse IgG negative control antibodies were serially diluted to concentrations of 10,000–0.419 ng/mL in 1% BSA and 0.05% Tween 20 in 1X PBS. Coated and blocked plates were washed, then incubated with 50 µL of each antibody dilution in duplicate wells. Following antibody sample incubation, plates were washed and incubated Goat anti-Mouse IgG—SulfoTag detection antibody in all wells. Following detection of antibody incubation, plates were read and the signal remained in direct correlation to antibody concentration, and curves for ^127^I-labeled and unlabeled anti-GFP and anti-mCherry antibodies were parallel through the linear range, indicating the ^127^I-labeled forms of each antibody maintained antigen-binding capability suitable for the intended application as SPECT/CT imaging agents.

### 2.9. PET/CT Imaging

Study subjects were randomly selected to equivalently balance average tumor volume across two cohorts of n = 3–4 subjects per group at 3- and 7-weeks post BxPc-3 tumor cell inoculation for in vivo [^18^F]-FDG or [^18^F]-FLT PET/CT imaging. Subjects were scanned using a multi-bed apparatus supporting n = 4 subjects/scan. Subjects were maintained under isoflurane anesthesia throughout the course of each scan. Subjects assigned to [^18^F]-FDG scanning cohorts were appropriately fasted, with free access to drinking water, prior to scanning to reduce image artifacts related to systemic metabolic activity in tissues surrounding the tumor site, and blood samples were collected for glucose monitoring. [^18^F]-FDG and [^18^F]-FLT were administered via the tail vein at a total radioactive dose level of approximately 200 uCi/subject; PET scanning was initiated at approximately 40 min post administration of the applicable imaging agent. CT images were acquired following each PET acquisition. CT-based attenuation, deadtime, random, and scatter corrections were applied to all data. Image data were analyzed using VivoQuant software version 3.1.4., InviCRO, LLC, Boston, MD, U.S.A. Fixed volume regions of interest (ROIs) were used to quantify total radioactivity in the tumor and heart for each animal at each time point; the results are reported as the standardized uptake value (SUV ((decay-corrected activity of tissue volume)/(injected activity/body mass)) for each ROI. ROIs for primary analysis included the tumor and heart; the heart was used as a surrogate to determine systemic activity. Due to previously reported inherent challenges associated with various in vivo imaging modalities, such as CT, MRI, PET, SPECT, etc., and data analysis with imaging related to PDAC and other forms of cancer with complex tumor microenvironments, [^18^F]-FDG PET data were also analyzed to determine total lesion glycolysis (TLG) via the analysis methodology as previously reported [99,100,101,102], with a thresholding value of 30%. An adaptation of TLG analysis was also applied to [^18^F]-FLT PET data sets, again with a 30% thresholding value and reported herein as total lesion proliferation (TLP). Our adaptation of TLG to determine TLP consisted of the substitution of [^18^F]-FDG metabolic tumor volume, which is a measure of tumor volume with high metabolic activity, with the [^18^F]-FLT-derived tumor volume, demonstrating high proliferative activity. We considered this the “proliferating tumor volume.”

### 2.10. SPECT/CT Imaging

Study subjects were randomly selected to equivalently balance average tumor volume within a single cohort of n = 3 subjects per group at 7–8 weeks post BxPc-3 tumor cell inoculation for in vivo [^125^I]-anti-GFP or [^125^I]-anti-mCherry SPECT/CT imaging. Subjects were scanned using a multi-bed apparatus supporting n = 3 subjects/scan. Subjects were maintained under isoflurane anesthesia throughout the course of each scan. [^125^I]-anti-GFP or [^125^I]-anti-mCherry was administered via the tail vein at a total radioactive dose level of approximately 200 uCi/subject; SPECT scanning was initiated at approximately 48 h post administration of the applicable imaging agent. CT images were acquired immediately following completion of SPECT imaging. Image data were analyzed using VivoQuant software version 3.1.4., InviCRO, LLC, Boston, MA, U.S.A. Fixed-volume regions of interest (ROIs) were used to quantify total radioactivity in the tumor and heart for each animal at each time point; the results are reported as the standardized uptake value (SUV), and the heart was used as a surrogate to determine systemic activity.

### 2.11. Quantitative Whole-Body Autoradiography (QWBA)

Following completion of SPECT/CT imaging, each subject receiving [^125^I]-anti-GFP or [^125^I]-anti-mCherry was further assessed for intratumoral and potential systemic expression of oncolytic virus transgene expression through the application of quantitative whole-body autoradiography (QWBA) via standard techniques. At least four quality control standards were placed in each frozen block of carboxy methyl cellulose (CMC) prior to sectioning and were used for section thickness quality control. Sections approximately 30 µm thick were taken from the sagittal plane and captured on adhesive tape. Appropriate sections selected at various levels of interest in the block were collected to encompass the required tissues and biological fluids where possible. The sections were exposed to phosphor imaging screens for approximately 48 h prior to digital scanning.

Quantification relative to the calibration standards was performed by image densitometry using MCID^TM^ image analysis software version 7.1., Imaging Research Inc., St. Catherines, ON, Canada. A standard curve was constructed, and a lower limit of quantification (LLOQ) was applied to the data. The LLOQ was determined to be 0.25–0.36 ng equivalents by using the radioactive concentration of the lowest calibration standard used to generate a calibration curve divided by the specific activity of the dose formulation (µCi/µg). Artifacts were excluded as necessary from the analysis during image processing.

### 2.12. CD-3^+^ T-Cell Biomarker Assay

To assess T-cell biomarker levels in vivo, baseline blood samples were collected from a random cohort of n = 4 subjects in each control and treatment group during week 2 and post-treatment blood samples were collected prior to in vivo imaging during weeks 7–8. Samples were analyzed via a multiplex panel for the following biomarkers, and the results are reported as picogram per milliliter (pg/mL): MCP-1, IL-2, IL-4, Il-6, IL-10, INF-γ, and TNF-α.

### 2.13. Tissue Processing and Histopathological Assessment

Following the final PET/CT or SPECT/CT, each subject was euthanized in compliance with Institutional Animal Care and Use Committee guidelines. Following euthanasia, each subject was examined carefully for external abnormalities, including palpable masses. The skin was reflected from a ventral midline incision, and any subcutaneous masses were identified and correlated with antemortem findings where applicable. Care was taken not to disrupt the tumor tissue associated with the primary xenograft implantation site and immediate adjacent tissue. Samples of the xenograft tumor (entire tumor, including the capsule) and skin adjacent to the tumor, brain, heart, kidney, liver, spleen, lung, and skeletal muscle were collected/processed for hematoxylin and eosin (H&E) staining. Samples of the xenograft tumor (entire tumor, including the capsule) and skin adjacent to the tumor were also collected and processed for immunohistochemistry (IHC) staining. Target tissues assigned to H&E processing were assessed by a Board-certified veterinary pathologist for, but not limited to, the following: overall size of the tumor section, nature and form of neoplastic cells, mitotic index, presence of dead or degrading cells, presence and character of inflammatory cells, invasion of inflammatory cells into the main body of the xenograft tissue, and fibrosis. Target tissues assigned for IHC staining were processed and stained according to established methods for the following biomarkers, with appropriate reagents as applicable for murine species: CD-3, CD-4, CD-8, CD-68, and caspase-3. Each immunohistochemical staining run contained proper positive and negative controls, and the controls were evaluated to ensure run validity and adequate inter-run consistency.

### 2.14. Statistical Analysis

For statistical analysis of body weight (g), QWBA (µCi/g tumor tissue), and SPECT/CT (SUV) datasets, an analysis of variance (ANOVA) with adjustment for multiple comparisons was performed. For endpoints and/or parameters (within each collection interval) where all groups with sample sizes of three or greater were included, the system was tested for the normality of the residuals and homogeneity of variances to see whether the data were approximately normal or whether a log transformation or rank transformation would be used. Leven’s test [103] was used to assess homogeneity of group variances and Shapiro–Wilk’s [104] test was used to test the normality of the residuals. A one-way analysis of variance using the appropriate transformed data was used to test each endpoint for the effects of treatment [105]. The results of these pair-wise comparisons were analyzed at the 0.05 and 0.01 significance levels after adjustment for multiple comparisons using the methods of Edwards and Berry [106]. All tests were two-tailed tests unless otherwise indicated.

For statistical analysis of tumor volume, cytokines, and PET/CT datasets, a repeated measures analysis of covariance (RMANCOVA) was performed. A repeated measures analysis of covariance (mixed model) was conducted for tumor volumes collected from measurement period 15 onwards; see Appendix A for the temporal relationship of measurement periods relative to study activities. For each endpoint, the model was tested for the effects of treatment, time, and the interaction of treatment and time [105]. Tumor volume data collected at termination of in-life for each subject were used as a covariant. The results of all pair-wise comparisons were analyzed at the 0.05 and 0.01 significance levels after adjustment for multiple comparisons using the methods of Edwards and Berry [106]. All endpoints were analyzed using two-tailed tests unless otherwise indicated.

## 3. Results

### 3.1. Replication Kinetics of TPV in Human Pancreatic Cancer Cells

Replication competency was assessed via in vitro assays and was confirmed for all TPV recombinants in BxPC-3 pancreatic cancer cells. Similarly, reporter gene expression was confirmed for all TPV recombinants via phase contrast and fluorescent microscopy, demonstrating eGFP or mCherry protein expression in association with viral plaque formation consistent with TPV-infected cells.

Replication efficiency of each TPV recombinant in BxPC-3 human PDAC cells was assessed via viral plaque assay, and the results demonstrated that the replication efficiency for all recombinants was comparable to published data for each TPV recombinant cultured in owl monkey kidney (OMK) cells, which serve as primary cell line for TPV production, as previously reported [81,82,83,94].

### 3.2. Treatment of BxPc-3 Xenografts with Tanapoxvirus Recombinants In Vivo

Tumor volume was measured throughout the course of the study in 3-day intervals to assess the potential treatment effect of the tanapoxvirus variant treatment compared to control treatment. Tumors in many subjects were noted to be tangibly multi-compartmental when physically palpated for tumor measurement. Multi-compartmental composition of tumor masses was also demonstrated through PET/CT imaging.. The results of tumor volume over time for each study cohort are shown in Figure 1 and Figure 2. When assessed in the immunocompromised BALB-c nude mouse model, each TPV recombinant demonstrated regression in BxPc-3 human PDAC cells compared to immunocompromised vehicle-treated control subjects. While TPV/eGFP and TPV/∆66R/m-IL-2/mCherry demonstrated a trend of regressed tumor volume over the course of the study, both recombinants failed to reach statistical significance (Figure 1A,B). However, TPV/∆66R/m-CCL-2/mCherry virotherapy achieved statistical significance (*p* < 0.05) at approximately 30 days post treatment; see Figure 1C. When assessed in the CD-3^+^ T-cell-mediated, immune-reconstituted, BALB-c nude mice, each TPV recombinant demonstrated relatively equivalent regression in BxPc-3 tumor volume compared to immune-reconstituted, vehicle-treated control subjects (Figure 2). TPV/∆66R/m-IL-2/mCherry and TPV/∆66R/m-CCL-2/mCherry virotherapy in immune-reconstituted animals demonstrated similar trends over the course of the study, with the results from both recombinants showing less overall regression in tumor volume compared to control (Figure 2B,C). TPV/∆66R/m-CCL-2/mCherry reached statistically greater mean tumor volume, with *p* < 0.05, compared to control at approximately 30 days post virotherapy (Figure 2A,C). However, TPV/eGFP recombinant-treated subjects demonstrated a relatively consistent tumor growth regression effect compared to control throughout the course of the study (Figure 2A). CD-3^+^ T-cell-mediated, immune-reconstituted control mice demonstrated statistically significant regressed tumor volume, with *p* < 0.01, compared to control immunocompromised mice at approximately 27 days post vehicle treatment (Figure 3).

### 3.3. Tumor Metabolic Activity Assessment via [^18^F]-FDG PET/CT Imaging

PET/CT imaging, in vivo [^18^F]-FDG PET/CT was used to assess BxPc-3 human PDAC tumor xenograft intratumoral metabolic activity in immunocompromised and CD-3^+^ T-cell-mediated immune-reconstituted mice. Images were analyzed to quantify total radioactivity within the region of the tumor as described herein; data are reported as standardized uptake value (SUV) and total lesion glycolysis (TLG) with 30% thresholding. Immunocompromised and immune-reconstituted group results are presented in Table 1 and Figure 4A (SUV) and Figure 4B (TLG), and Figure 5A (SUV) and Figure 5B (TLG), respectively. Our results fail to demonstrate statistical difference within and between each group in both the immunocompromised and the immune-reconstituted subjects assessed. Furthermore, total radioactivity within the region of the tumor, whether analyzed to determine SUV or TLG, for many treatment conditions did not correlate with final group mean tumor volume, particularly with respect to the week 7–8 final group mean tumor volume results. Select representative post-virotherapy [^18^F]-FDG PET/CT images are shown in Figure 6 for vehicle-treated and TPV-eGFP-treated immunocompromised and immune-reconstituted mice. 

### 3.4. Tumor Cell Proliferation Assessment via [^18^F]-FLT PET/CT Imaging

PET/CT imaging in vivo [^18^F]-FLT PET/CT was used to assess BxPc-3 human PDAC tumor xenograft intratumoral cell proliferation activity. Images were analyzed to quantify total radioactivity within the region of the tumor as described herein; data are reported as SUV and total lesion proliferation (TLP) with 30% thresholding. Immunocompromised and immune-reconstituted group results are presented in Table 2 and demonstrated in Figure 7A (SUV) and Figure 7B (TLP), and Figure 8A (SUV) and Figure 8B (TLP), respectively. [^18^F]-FLT SUV results failed to demonstrate statistical difference within and between each study group in both the immunocompromised and the immune-reconstituted mice, and consistent with the [^18^F]-FDG SUV results, total radioactivity within the region of the tumor did not correlate with week 7–8 tumor volume data. As seen with the [^18^F]-FDG SUV results, [^18^F]-FLT TLP vehicle-treated, intra- and intergroup comparisons for all TPV variant-treated immunocompromised and immune-reconstituted study groups failed to achieve statistically significant differences. However, a trend was seen when comparing baseline to post-virotherapy [^18^F]-FLT TLP results in the immunocompromised and immune-reconstituted study groups that also correlates to final total tumor volume. The [^18^F]-FLT TLP vehicle-treated, immunocompromised mice results demonstrated a statistically significant difference, *p* < 0.05, between baseline and post virotherapy. Likewise, the post-treatment vehicle-treated and TPV/∆66R/mCCL-2-treated immune-reconstituted study groups demonstrated a statistically significant difference, *p* < 0.05, when compared to post-virotherapy immune-reconstituted study subjects. Select representative post-virotherapy [^18^F]-FLT PET/CT images are shown in Figure 9 for vehicle-treated and TPV-eGFP-treated immunocompromised and immune-reconstituted mice.

### 3.5. In Vivo Tumor TPV Transgene Expression Assessment via SPECT/CT Imaging

SPECT/CT imaging, in vivo [^125I^]-anti-eGFP and [^125I^]-anti-mCherry antibody SPECT/CT imaging, was used to assess TPV-variant transgene expression in BxPc-3 human PDAC tumor xenografts in immunocompromised and immune-reconstituted BALB-c nude mice as described herein. Post-virotherapy images were analyzed to quantify percent injected dose per gram tissue (%ID/g) of total administered radioactivity per subject within the region of the tumor; group means were calculated and plotted against tumor volume. Immunocompromised and immune-reconstituted group results are presented in Table 3 and demonstrated in Appendix A, respectively: immunocompromised and immune-reconstituted group mean tumor volume and [^125^I]-anti-eGFP or [^125^I]-anti-mCherry antibody SPECT/CT percent injected dose per gram of tissue.

The results demonstrate relatively consistent %ID/g values across all immunocompromised groups, including the vehicle-treated control group, and do not demonstrate tumor-volume dependency; mean %ID/g ± SEM for all immunocompromised study groups = 2.42 ± 0.46. Likewise, the results for immune reconstituted are also relatively consistent, again including the vehicle-treated control group, and also do not demonstrate tumor-volume dependency, although the mean was slightly lower than that for immunocompromised subjects; mean %ID/g ± SEM for all immune reconstituted study groups = 1.48 ± 0.18. Select representative SPECT/CT images are shown in Figure 10 for vehicle-treated and TPV-eGFP-treated immunocompromised and immune-reconstituted subjects. A common subjective finding through review of PET/CT and SPECT/CT images is the association of all imaging agents with the superficial aspects of many of the tumor masses. This observation demonstrates a lack of consistent penetration of the imaging agents into the core of many tumors (Figure 6A,C, Figure 9A and Figure 10A).

### 3.6. Ex Vivo Tumor TPV Transgene Expression Assessment via Quantitative Whole-Body Autoradiography (QWBA)

TPV transgene expression was assessed following completion of SPECT/CT imaging through application of quantitative whole-body autoradiography (QWBA). Images were analyzed to quantify nanogram antibody per gram tissue (ng-Ab/g) per animal within the region of the tumor as described herein. Immunocompromised and immune-reconstituted group results are presented in Table 4 and demonstrated in Appendix A and S2B, respectively: immunocompromised and immune-reconstituted [^125^I]-anti-eGFP or [^125^I]-anti-mCherry antibody quantitative whole-body autoradiography nanogram-antibody per gram of tissue. The results demonstrate treatment-independent, tumor volume-dependent results in immunocompromised and immune-reconstituted study groups. Select representative QWBA images are shown in Appendix A: [^125^I]anti-eGFP or [^125^I]anti-mCherry antibody quantitative whole-body autoradiography images of BxPc-3 PDAC human tumor xenografts in immunocompromised and immune-reconstituted BALB-c nude mice, for vehicle-treated and TPV-eGFP-treated immunocompromised and immune-reconstituted subjects.

### 3.7. Biomarker Analysis and Pathology

To assess T-cell biomarker levels in vivo, baseline and post-virotherapy blood samples were collected and analyzed as a multiplex panel for the following biomarkers: MCP-1, IL-2, IL-4, Il-6, IL-10, INF-γ, and TNF-α. The baseline sample results were unremarkable; all samples collected from each study group for each biomarker were below the limit of quantitation (BLQ) of the multiplex assay, except for IL-10 (maximum level = 13.8 pg/mL), IL-6 (maximum level = 30.1 pg/mL), and MCP-1 (maximum level = 48.1 pg/mL). The post-virotherapy sample assay results for all immunocompromised study groups were also BLQ for all biomarkers assayed, except for the following: vehicle (IL-6 = 19.1 pg/mL), TPV/eGFP (IL-10 = 10.9 pg/mL, IL-6 = 16.2 pg/mL, TNF-α = 530.1 pg/mL, and MCP-1 = 451.9 pg/mL), TPV/∆66R/mCCL-2 (IL-6 = 478.8 pg/mL), and TPV/∆66R/mIL-2 (INF-γ = 25.1 pg/mL and IL-6 = 18.2 pg/mL). The post-virotherapy sample assay results for immune reconstituted study groups are summarized as follows: vehicle (IL-6 = 1951 pg/mL), TPV/eGFP (IL-10 = 108.8 pg/mL, IL-6 = 60.8 pg/mL, TNF-α = 13.1 pg/mL, and MCP-1 = 60.3 pg/mL), TPV/∆66R/mCCL-2 (IL-10 = 12.5 pg/mL, IL-6 = 37.4 pg/mL), and TPV/∆66R/mIL-2 (IL-10 = 23.8 pg/mL). The results for immunocompromised study groups failed to demonstrate trends when compared between baseline to post-virotherapy samples or when compared between study groups. Likewise, the results for immune reconstituted study groups also failed to demonstrate a clear trend when compared between baseline to post-virotherapy samples or when compared between study groups. Adoptive CD-3^+^ T-cell transfer does not appear to have resulted in a T-cell-dependent increase in biomarker levels at the timepoint sampled in our study (week 7–8).

### 3.8. Pathology

Ex vivo histopathological assessments were conducted to determine the potential treatment effect in all study subjects. Microscopic examination of fixed H&E-stained paraffin sections was performed on tissue samples of the xenograft tumor and immediately adjacent tissue from a cohort of n = 5 subjects from all immunocompromised and immune-reconstituted study groups. Tissues from all study groups were also assessed via standard IHC staining techniques for the following biomarkers: CD-3, CD-4, CD-8, CD-68, and caspase-3; all tissues were examined by a board-certified veterinary pathologist. A five-step grading system was utilized to define gradable lesions for comparison between dose groups.

Macroscopic observations demonstrate that a mass or nodule occurred at the transplant sight of all animals in the immunocompromised and immune-reconstituted groups. These correlated with adenocarcinoma or transplant necrosis noted microscopically. There were no other meaningful macroscopic findings. The other macroscopic findings noted in study subjects were of low incidence and had no treatment-related microscopic correlate.

Microscopic results for immunocompromised and immune-reconstituted groups are presented in summary results in Table 5 and Table 6, respectively, and key microscopic results are presented in the text immediately below. Select representative histopathology and immunohistochemistry photomicrographs are presented in Figure 11.

All immunocompromised vehicle control subjects had tumor transplants (adenocarcinomas) in the subcutaneous tissue. The cells in these tumors were markedly caspase positive. Two of five immune-reconstituted vehicle control subjects demonstrated minimal staining of small numbers of tumor cells remaining in the tumor transplant. The remaining subjects in the immune-reconstituted vehicle control group were caspase negative, as tumor cells were not present, although an increase in CD-3, CD-4, and CD-68 staining was observed in the tumor transplant area. This staining was principally in the inflammation that surrounded the replaced tumor cells. Thus, increased T-cells (CD-3 and CD-4 positive cells) and macrophages (CD-68 positive cells) occurred in immune-reconstituted mice. All immunocompromised TPV/eGFP subjects had tumor transplants (adenocarcinomas) in the subcutaneous tissue. Two of six immune-reconstituted TPV/eGFP subjects had adenocarcinomas, and four of six mice had caspase positivity that ranged from minimal in two subjects with only minor residual tumor cells to moderate and marked in two other mice, with much less tumor impact. Increased CD-3, CD-4, and CD-68 staining occurred in the tumor transplant area around the regions of tumor necrosis in immune-reconstituted TPV/eGFP subjects when compared to immunocompromised TPV/eGFP-treated mice. This staining was principally in the inflammation that surrounded the replaced tumor cells. Thus, increased T-cells (CD-3- and CD-4-positive cells) and macrophages (CD-68-positive cells) occurred in immune-reconstituted mice. All immunocompromised TPV/∆66R/mCCL-2 mice had tumor transplants (adenocarcinomas) in the subcutaneous tissue. All immune-reconstituted TPV/∆66R/mCCL-2 subjects lacked an identifiable tumor and were caspase negative, reflecting the absence of tumor cells in the tumor transplant area. CD-3, CD-4, and CD-68 staining was comparable for most immunocompromised TPV/∆66R/mCCL-2 and immune-reconstituted TPV/∆66R/mCCL-2 subjects, with no clear distinction of staining between these groups. Chronic–active inflammation was of increased severity in immune-reconstituted TPV/∆66R/mCCL-2 mice (marked) compared to mild chronic–active inflammation in immunocompromised TPV/∆66R/mCCL-2 subjects. Tumor necrosis was complete (severe) in all immune-reconstituted TPV/∆66R/mCCL-2 mice. Partial necrosis (minimal to moderate) occurred in most immunocompromised TPV/∆66R/mCCL-2 subjects and generally only involved a portion of the tumor. All immunocompromised TPV/∆66R/mIL-2 subjects had tumor transplants (adenocarcinomas) in the subcutaneous tissue. Five of six immune-reconstituted TPV/∆66R/mIL-2 mice had no identified tumor and were caspase negative, reflecting the absence of tumor cells in the tumor transplant area. A single immune-reconstituted TPV/∆66R/mIL-2 mouse had a tumor transplant (adenocarcinoma) at the transplant site, which indicates the subject did not respond to adoptive CD-3^+^ T-cell transfer, did not respond to virotherapy, or that either the T-cell or TPV/∆66R/mIL-2 virotherapy administrations were potentially partial or missed administrations. Increased CD-3 and CD-4 staining occurred in the tumor transplant area around the regions of tumor necrosis in immunocompromised TPV/∆66R/mIL-2 mice compared to immune-reconstituted TPV/∆66R/mIL-2 mice. This staining was principally in the inflammation that surrounded the replaced tumor cells. CD-68 staining was slightly more pronounced for immune-reconstituted TPV/∆66R/mIL-2 mice compared to immunocompromised TPV/∆66R/mIL-2 mice. Thus, increased T-cells (CD-3- and CD-4-positive cells) and macrophages (CD-68-positive cells) occurred in immune-reconstituted TPV/∆66R/mIL-2 mice. Chronic–active inflammation was of increased severity in immune-reconstituted TPV/∆66R/mIL-2 mice (marked) compared to minimal to moderate chronic–active inflammation in immunocompromised TPV/∆66R/mIL-2 mice. Necrosis was complete (severe) in all animals of immune-reconstituted TPV/∆66R/mIL-2 mice that had a response to treatment. Partial necrosis (minimal to moderate) occurred in most immunocompromised TPV/∆66R/mIL-2 mice and generally only involved a portion of the tumor. Chronic active inflammation was comparable across the TPV virotherapeutics in immunocompromised mice and was greater than in immunocompromised vehicle controls. Thus, TPV virotherapy-treated groups demonstrated an increase in chronic active inflammation and necrosis relative to immunocompromised vehicle control subjects.

## 4. Discussion

In this study, we evaluated the immuno-oncolytic potential of various TPV variants against BxPc-3 human PDAC xenografts in immunocompromised and CD-3^+^ T-cell-mediated immune-reconstituted BALB-c nude mice through application of a rigorous in vitro, in vivo, and ex vivo experimental approach. Our in vitro results clearly demonstrate that TPV/eGFP, TPV/∆66R/mCCL-2, and TPV/∆66R/mIL-2 could replicate efficiently, infect, cause morphological changes consistent with TPV-dependent cytopathic effect, cause cell lysis, and express viral transgene expression in BxPc-3 PDAC cells. TPV/eGFP replication efficiency was shown to be relatively equivalent in BxPc-3 cells compared to owl monkey kidney cells traditionally used for in vitro TPV culture [81,82,83,94].

Our in vivo results in immunocompromised mice demonstrate that TPV recombinants expressing mCCL-2 and mIL-2 were more effective in the regression in BxPc-3 PDAC tumor volume compared to TPV/eGFP and vehicle control. The TPV/∆66R/mCCL-2 group mean final tumor volume was 193.1 mm^3^ less than the group mean final tumor volume of the vehicle control group and achieved statistical significance, with *p* < 0.05, in immunocompromised athymic nude mice. As observed with TPV/∆66R/mCCL-2, TPV/∆66R/mIL-2 demonstrated a similar trend with respect to regression in tumor volume compared to TPV/eGFP- and vehicle-treated controls (final tumor volume difference of 128.2 mm^3^ between the mIL-2-treated and vehicle control groups) yet failed to reach a statistical difference throughout the course of the study.

The role of CCL-2 in the progression of pancreatic cancer has been of high interest for investigators given its potent induction of monocyte migration to the tumor site, and the production of CCL-2 correlates with macrophage presence in transplanted in vivo tumors [107,108,109]. CCL-2 also demonstrates chemoattractant properties for several other immune cells, such as NK cells, mast cells, monocytes, and T lymphocytes [85,86,87]. Investigators have also reported that CCL-2 may be an important negative regulator of pancreatic cancer progression [90]. IL-2 has also been shown to share similar immunomodulatory properties to those reported for CCL-2. IL-2 promotes immune response through the activation, differentiation, and maturation of T-cells, as well as immune response mediated through the activation of macrophages and NK cells [91,92]. Our histopathology and immunohistochemistry results from immunocompromised TPV/∆66R/mCCL-2 and TPV/∆66R/mIL-2 are also consistent with reported immunomodulatory properties of CCL-2 and IL-2. All TPV virotherapy-treated immunocompromised groups demonstrated an increase in chronic active inflammation and necrosis relative to immunocompromised vehicle control subjects. For all TPV recombinants, inflammatory biomarkers and immune cell infiltrates in and around the tumor sites were found to be greater than those of immunocompromised vehicle control subjects. TPV/∆66R/mCCL-2 treatment was shown to result in the greatest difference compared to control, followed by TPV/∆66R/mIL-2 and finally TPV/eGFP. Evidence of in vivo TPV variant(s) replication and transgene expression was clearly demonstrated through immunohistochemistry assessment of tumor tissues where an intact tumor was present upon termination of the study. Our results further support the potential role of CCL-2 and IL-2 in negative regulation of pancreatic tumor progression through stimulation of innate immune response given the regression in tumor volume within TPV/∆66R/mIL-2-treated subjects and statically significant regression in tumor volume in TPV/∆66R/mCCL-2-treated subjects compared to vehicle control immunocompromised subjects. TPV/eGFP-treated subjects demonstrated a marginal regression in tumor volume compared to vehicle controls, again supporting the role CCL-2 and IL-2 play in the negative regulation of BxPC-3 human PDAC xenografts in our immunocompromised test system.

All treatment groups, including vehicle control, of the immune-reconstituted mice demonstrated a regression in tumor volume compared to immunocompetent vehicle control animals. Nearly all immune-reconstituted mice had complete or near-complete necrosis of the tumor mass. The necrosis of the tumor in these immune-reconstituted mice was surrounded by a border of chronic inactive or chronic active inflammation. Inflammation was characterized in general as an accumulation of mixed inflammatory cells, including neutrophils, small mononuclear cells, large mononuclear cells (macrophages), and fibrosis surrounding a central core of necrotic debris. The tumor transplant cells were distinctly caspase positive in subjects where tumor necrosis was incomplete. Our results demonstrate that all TPV virotherapy-treated, immune-reconstituted mice failed to demonstrate a beneficial treatment effect compared to vehicle-treated, immune-reconstituted control mice. The TPV/eGFP-treated, immune-reconstituted group results trended most similarly to the immune-reconstituted, vehicle-treated control group results. While the TPV/∆66R/mCCL-2- and TPV/∆66R/mIL-2-treated, immune-reconstituted group results demonstrate a regression in tumor volume, the rate of regression and overall efficacy of both variants were found to be less effective compared to the TPV/eGFP and vehicle control groups. Given our in vivo tumor assessment results when considered in totality with the ex vivo histopathology and immunohistochemistry results, we hypothesize that the results observed in immune-reconstituted mice, independent of treatment, are indicative of the host’s innate immune system rejection of the human PDAC xenograft employed in our study. Our results demonstrate the temporal relationship between CD-3^+^ T-cell transfer to the start of tumor volume regression, the rapid rate of tumor volume regression, and the pathology results, collectively support host immune-mediated rejection of the xenograft, as our results are consistent with host immune system-mediated transplant rejection, as reported by many investigators [110,111,112,113,114]. The results demonstrating TPV/eGFP treatment were more similar to vehicle-treated, immune-reconstituted animals compared to the CCL-2 and IL-2 variants, with the thymidine kinase (TPV gene *66R*) knockout, likely being a result of maintained TPV/eGFP variant thymidine kinase gene activity. Furthermore, we hypothesize that the lessened rate of tumor regression observed for CCL-2 and IL-2 variants was also likely a result of immune system competition between clearance of foreign xenograft cells and TPV variants due to CCL-2- and IL-2-mediated immune cell recruitment and activation in combination with slower viral growth kinetics for the CCL-2 and IL-2 variants due to a lack of thymidine kinase activity. Further refinement of CD-3^+^ T-cell transfer as a means of providing an immune-reconstituted research model is clearly needed given the results we describe herein. Potential modifications include a reduction in the total number of transferred T-cells per subject, a requirement for a greater tumor volume at the time of T-cell transfer, and an extended temporal relationship between the time of oncolytic virotherapy and the time of T-cell transfer.

In vivo imaging results with [^18^F]-FDG, SUV, and TLG; SPECT/CT imaging results with [^125^I]-anti-GFP and [^125^I]-anti-mCherry antibodies; and ex vivo QWBA results were unremarkable/confounding and do not logically align with tumor volume or histopathology, specifically when evaluating week 7–8 data sets. This lack of alignment is exemplified when one compares the results of the [^18^F]-FDG PET/CT image data generated from immunocompromised vehicle control subjects to immunocompromised TPV/eGFP-, TPV/∆66R/mCCL-2-, and TPV/∆66R/mIL-2-treated subjects. The final group mean tumor volume for immunocompromised vehicle control subjects was greater than the final group mean tumor volume for all immunocompromised TPV variant-treated subjects, yet quantitative [^18^F]-FDG results were shown to be greater for all TPV-variant-treated subjects compared to vehicle control. SPECT/CT imaging results also do not align with tumor volume and/or applied treatments. This is most clearly demonstrated through assessment of the vehicle-treated control groups in both immunocompromised and immune-reconstituted models compared to any respective TPV virotherapeutic group in both models. Given that we employed a [^125^I]-radiolabeled, monoclonal antibody targeting GFP or mCherry, one would expect radioactivity levels in and around the tumor site in all control animals to be at or equivalent to the blood pool given the lack of viral transgene expression and target antigen availability; our results show similar levels of total radioactivity at the tumor sites for both the immunocompromised and the immune-reconstituted vehicle-treated control study groups compared to all TPV variant-treated study groups. We hypothesize that this phenomenon is likely attributable to the complex microenvironment associated with PDAC tumor sites, which results in significant challenges when employing in vivo imaging approaches to assess contemporaneous tumor characteristics, as previously reported [31,39,40,41]. Following primary analysis of [^18^F]-FDG and [^18^F]-FLT PET/CT data, the challenges as described immediately above were noted and the data further assessed to determine total lesion glycolysis (TLG) for [^18^F]-FDG data and total lesion proliferation (TLP) for [^18^F]-FLT data. TLG assessment for [^18^F]-FDG data yielded similar, confounding results, which do not align with the other end points or datasets generated through the course of our study. However, [^18^F]-FLT TLP results do trend logically with tumor volume and histopathology data. While [^18^F]-FLT TLP data did not reach statistical significance for any study group, the trends observed demonstrate that group mean [^18^F]-FLT TLP signal intensity levels correlate well with final group mean tumor volumes and histopathology results. [^18^F]-FLT TLP results demonstrate the TPV mCCL-2 recombinant OV to have been the most effective of the TPV recombinants assessed in this study. Briefly, [^18^F]-FLT PET results of the TPV mCCL-2 recombinant-treated group demonstrate consistently lower signal within the region of the tumor compared TPV/eGFP and vehicle control, which trended with final group mean tumor volume for each TPV variant in the immunocompromised mouse model.

While this method of assessing [^18^F]-FLT data has not been previously reported, to the best of our knowledge, we propose that this approach, though further assessment is clearly needed, may be a viable approach in the assessment of pancreatic cancer tumor staging and diagnosis via in vivo PET/CT imaging.

In conclusion, our results demonstrate that TPV recombinants TPV/∆66R/mIL-2 and TPV/∆66R/mCCL-2 were more effective in regression in BxPc-3 pancreatic tumor volume and tumor proliferation rate compared to TPV/eGFP control recombinant, with the mCCL-2 recombinant reaching statistical significance (*p* < 0.05) compared to vehicle-treated control animals in an immunocompromised athymic nude murine model. PET/CT imaging results using [^18^F]-FLT targeting cells with active proliferation and histopathologic analysis also support the TPV mCCL-2 recombinant to have been the most effective of the TPV recombinants assessed in this study. The results generated via [^18^F]-FLT PET demonstrate a consistently lower signal within the region of the tumor compared to vehicle controls, and the histology results demonstrate that increased CD3 and CD4 staining occurred in TPV mCCL-2 recombinant relative to other virus variants and vehicle control. The enhanced immuno-oncolytic virus efficacy of TPV/∆66R/mIL-2 and TPV/∆66R/mCCL-2 may be attributable to CCL-2- and IL-2-mediated negative regulation of pancreatic tumor progression through stimulation of the innate immune response. TPV appears to be a potentially viable immuno-oncolytic virus that can infect, replicate, and induce a cytopathic effect, leading to direct cell lysis and a regression in BxPc-3 human PDAC xenografts in vivo through induction of anti-tumor innate immune response.

## Figures and Tables

**Figure 1 biomedicines-12-01834-f001:**
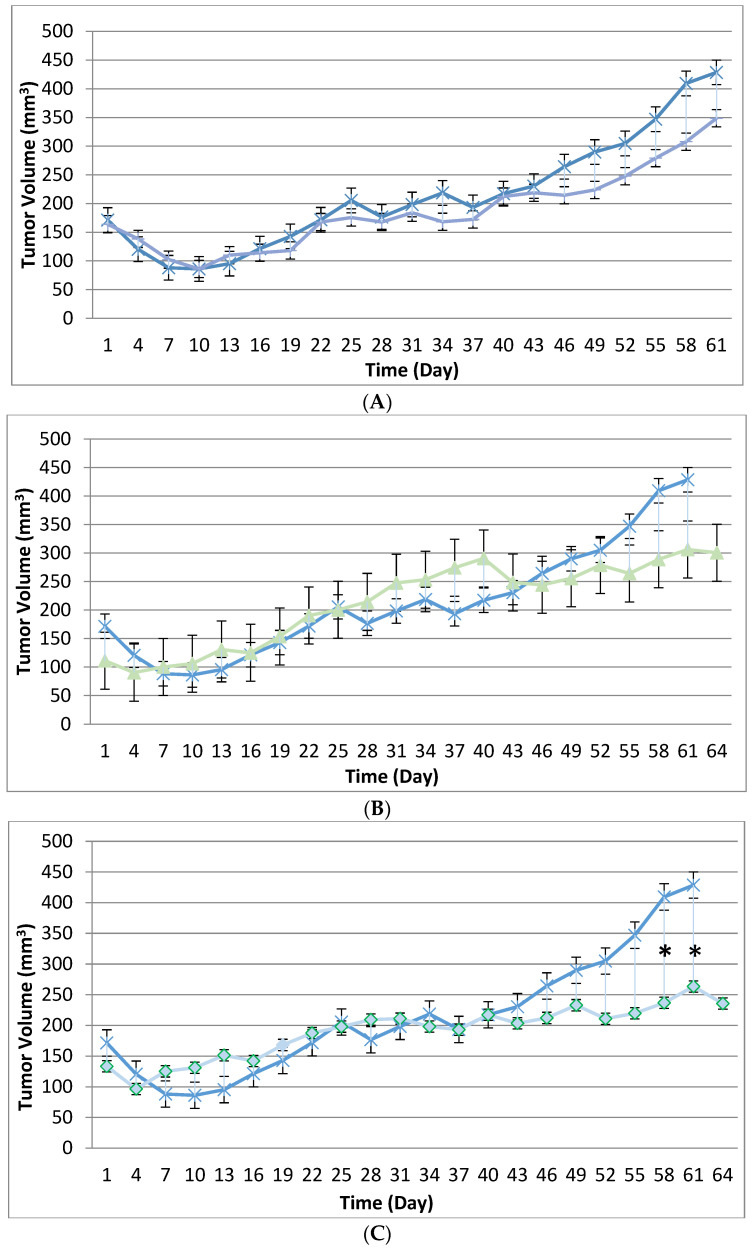
Immunocompromised group mean tumor volume. BxPc-3 human PDAC cells were inoculated subcutaneously (SC) in the right flank region of immunocompromised female BALB-c nude mice. Each subject received a single inoculation of 5 × 10^6^ cells/subject. Total volume measurement was conducted every three days ((length) × (width) × (height) × (П/6)). When tumors reached approximately 200 mm^3^, approximately 27 days post inoculation, animals were randomly assigned to study treatment groups based on tumor volume to achieve an equivalent average group tumor volume as much as possible. Each subject was treated with a single intratumoral administration of 100 µL PBS formulation vehicle or 100X virus stock of each applicable recombinant virus to result in a virus mass dose of 5 × 10^6^ plaque-forming units (pfu) per animal. Vehicle control average tumor volume is shown in each plot as x symbol plot line (blue) and used as a comparator for each TPV recombinant experimental group: TPV/eGFP shown as hyphen symbol plot line (purple) (**A**), TPV/∆66R/m-IL-2/mCherry shown as triangle symbol plot line (green) (**B**), and TPV/∆66R/m-CCL-2/mCherry shown as circle symbol plot line (green) (**C**). Bars show standard error of the mean (±1 SEM) and, where applicable, an asterisk (*) indicates statistically significant regression in TPV/∆66R/m-CCL-2/mCherry recombinant from vehicle control (*p* < 0.05).

**Figure 2 biomedicines-12-01834-f002:**
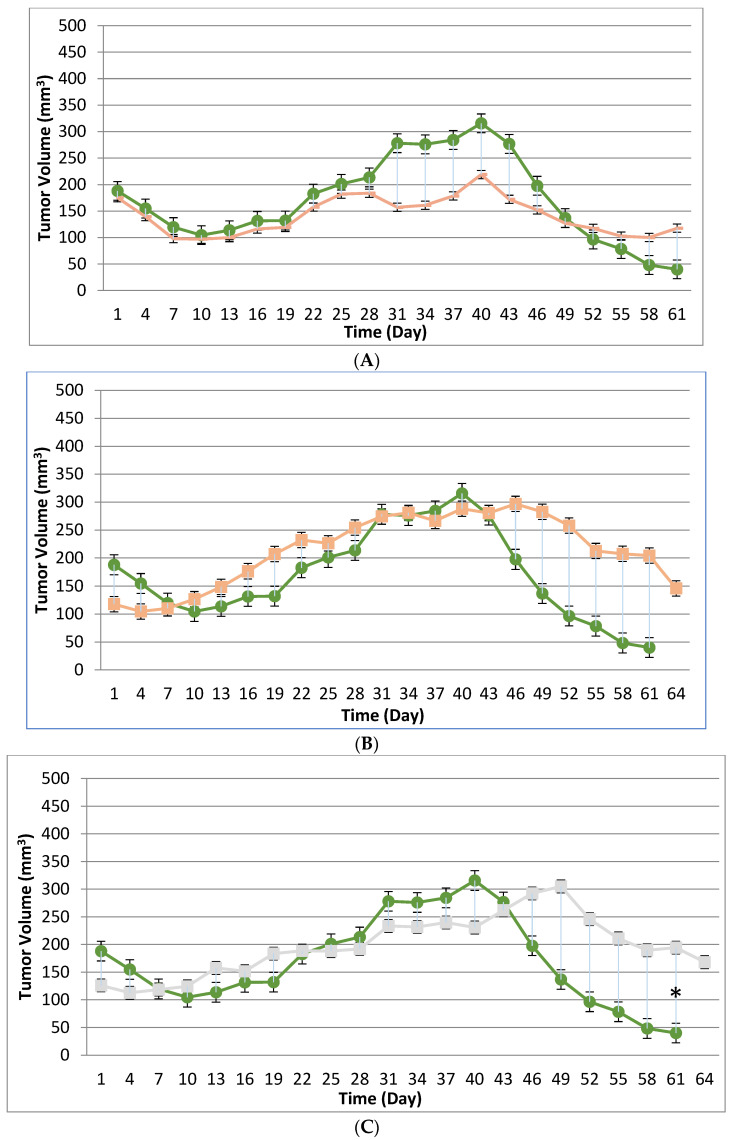
Immune-reconstituted group mean tumor volume. BxPc-3 human PDAC cells were inoculated subcutaneously (SC) in the right flank region of immune-reconstituted female BALB-c nude mice. Each subject received a single inoculation of 5 × 10^6^ cells/subject. Total volume measurement was conducted every three days ((length) × (width) × (height) × (П/6)). When tumors reached approximately 200 mm^3^, approximately 27 days post inoculation, animals were randomly assigned to study treatment groups based on tumor volume to achieve an equivalent average group tumor volume as much as possible. Each subject was treated with a single, intratumoral administration of 100 µL PBS formulation vehicle or 100X virus stock of each applicable recombinant virus to result in a virus mass dose of 5 × 10^6^ plaque-forming units (pfu) per animal. Four days post vehicle control or virotherapy, each subject received a CD-3+ T-cell adoptive cell transfer (5 × 10^6^ cells/subject) via tail vein administration. The vehicle control average tumor volume is shown in each plot as circle symbol plot line (green), and used as a comparator for each TPV recombinant experimental group: TPV/eGFP shown as hyphen symbol plot line (peach) (**A**), TPV/∆66R/m-IL-2/mCherry shown as square symbol (peach) (**B**), and TPV/∆66R/m-CCL-2/mCherry shown as square symbol (grey) (**C**). Bars show standard error of the mean (±1 SEM) and, where applicable, an asterisk (*) indicates a statistically significant greater mean group tumor volume of TPV/∆66R/m-CCL-2/mCherry recombinant from vehicle control (*p* < 0.05).

**Figure 3 biomedicines-12-01834-f003:**
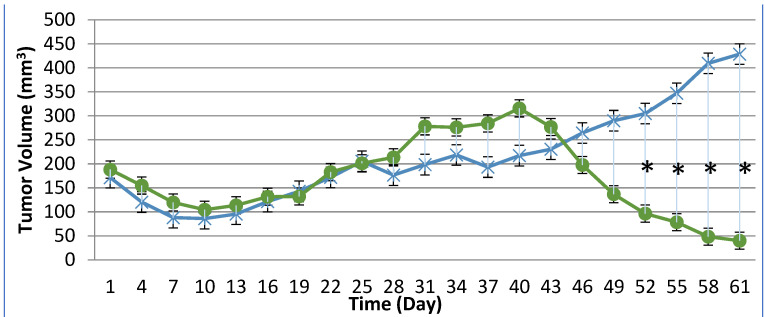
Immunocompromised and immune-reconstituted vehicle control group mean tumor volume. BxPc-3 human PDAC cells were inoculated subcutaneously (SC) in the right flank region of immunocompromised and immune-reconstituted female BALB-c nude mice. Each subject received a single inoculation of 5 × 10^6^ cells/subject. Total volume measurement was conducted every three days ((length) × (width) × (height) × (П/6)). When tumors reached approximately 200 mm^3^, approximately 27 days post inoculation, animals were randomly assigned to study treatment groups based on tumor volume to achieve an equivalent average group tumor volume as much as possible. Each subject was treated with a single intratumoral administration of 100 µL PBS formulation vehicle. Four days post vehicle control, each immune-reconstituted subject received a CD-3+ T-cell adoptive cell transfer (5 × 10^6^ cells/subject) via tail vein administration. Immunocompromised vehicle control average tumor volume is shown in the plot as x symbol plot line (blue) and immune-reconstituted, vehicle control average tumor volume is displayed as circle symbol (green) plot line. Bars show standard error of the mean (±1 SEM) and, where applicable, an asterisk (*) indicates a statistically significant difference in mean group tumor volume between the vehicle-treated control groups (*p* < 0.01).

**Figure 4 biomedicines-12-01834-f004:**
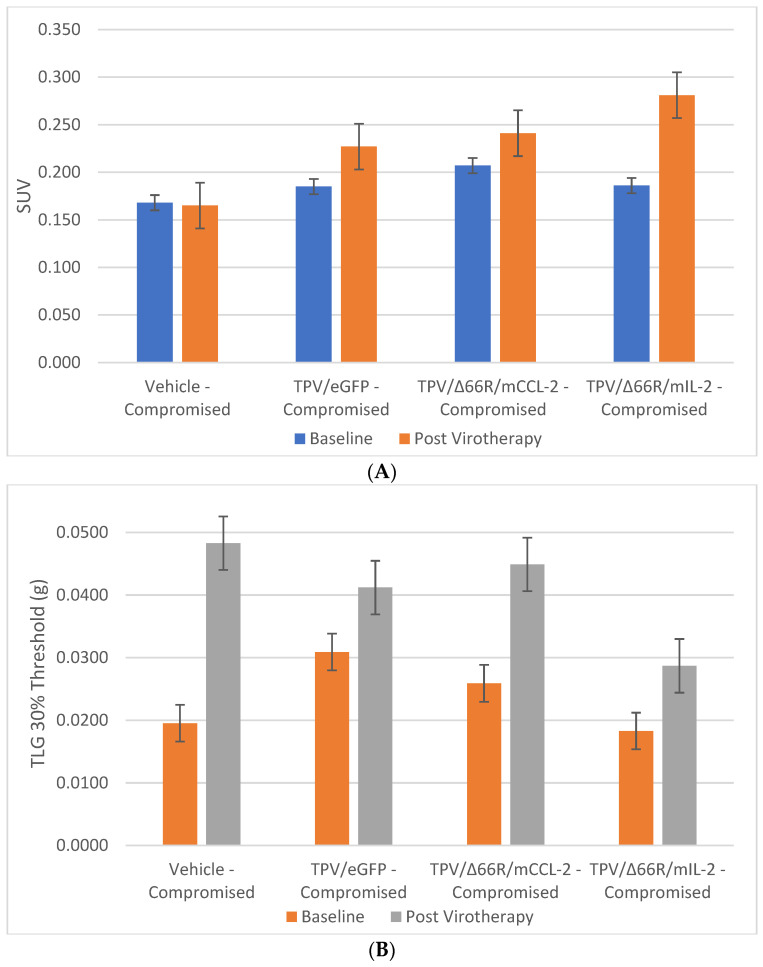
Immunocompromised group mean tumor [^18^F]-FDG PET/CT-derived standardized uptake value (SUV) and total lesion glycolysis (TLG). BxPc-3 human PDAC cells were inoculated subcutaneously (SC) in the right flank region of immunocompetent female BALB-c nude mice. Each subject received a single inoculation of 5 × 10^6^ cells/subject. Each subject was treated with a single intratumoral administration of 100 µL PBS formulation vehicle or 100X virus stock of each applicable recombinant virus to result in a virus mass dose of 5 × 10^6^ plaque-forming units (pfu) per animal when tumors reached approximately 200 mm^3^. Baseline [^18^F]-FDG PET/CT images (200 µCi/subject) prior to therapy were acquired 3 weeks post BxPc-3 cell inoculation, and post-virotherapy images were collected at 7–8 weeks post inoculation. Bars show standard error of the mean (±1 SEM); SUV results for each respective group are shown in plot (**A**) and TLG results are show in plot (**B**); inter- and intragroup statistical comparisons resulted in a lack of statistical significance for all comparisons.

**Figure 5 biomedicines-12-01834-f005:**
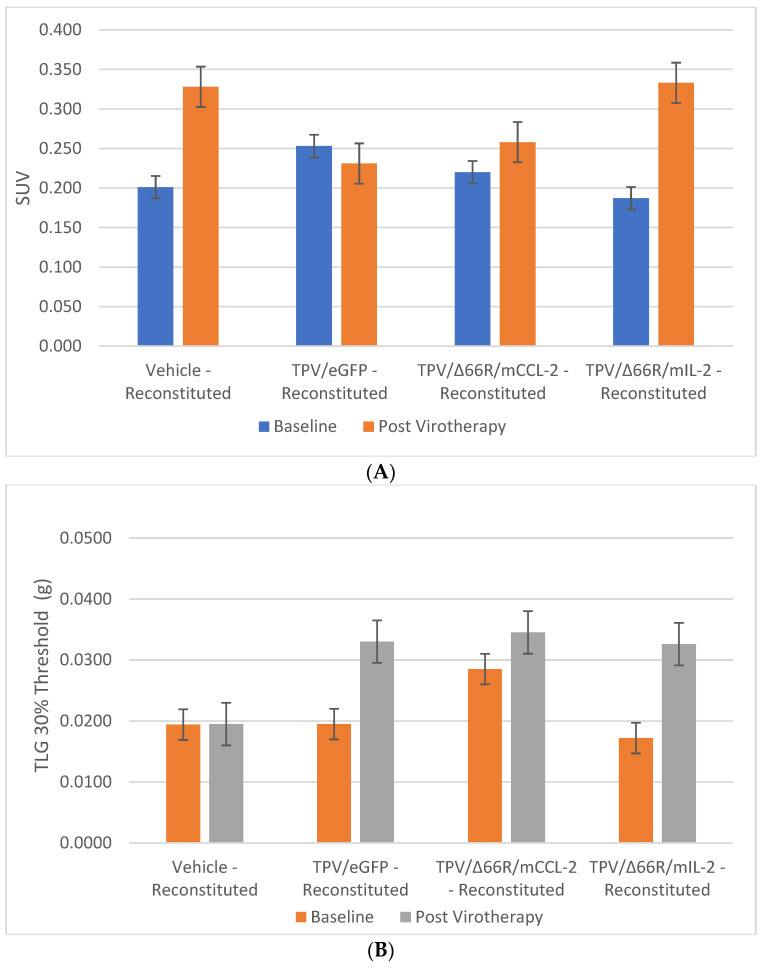
Immune-reconstituted group mean tumor [^18^F]-FDG PET/CT-derived standardized uptake value (SUV) and total lesion glycolysis (TLG). BxPc-3 human PDAC cells were inoculated subcutaneously (SC) in the right flank region of immunocompromised or immune-reconstituted female BALB-c nude mice. Each subject received a single inoculation of 5 × 10^6^ cells/subject. Each subject was treated with a single intratumoral administration of 100 µL PBS formulation vehicle or 100X virus stock of each applicable recombinant virus to result in a virus mass dose of 5 × 10^6^ plaque-forming units (pfu) per animal when tumors reached approximately 200 mm^3^. Baseline [^18^F]-FDG PET/CT images (200 µCi/subject) prior to virotherapy were acquired 3 weeks post BxPc-3 cell inoculation, and post-virotherapy images were collected at 7–8 weeks post inoculation. Bars show standard error of the mean (±1 SEM); SUV results for each respective group are shown in plot (**A**) and TLG results are show in plot (**B**); inter- and intragroup statistical comparisons resulted in a lack of statistical significance for all comparisons.

**Figure 6 biomedicines-12-01834-f006:**
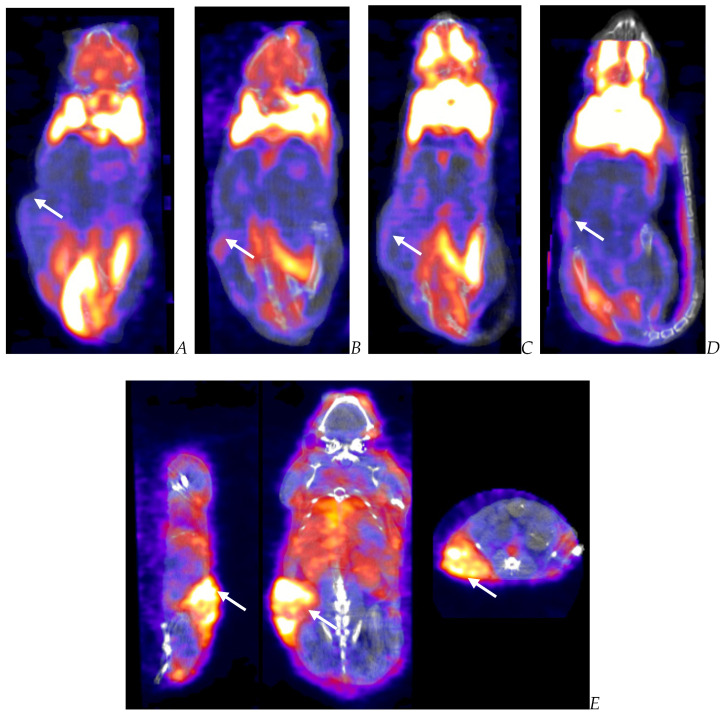
Post-virotherapy [^18^F]-FDG PET/CT Images of BxPc-3 PDAC human tumor xenografts in immunocompromised and immune-reconstituted BALB-c nude mice. BxPc-3 human PDAC cells (5 × 10^6^ cells/subject) were inoculated subcutaneously (SC) in the right flank region of immunocompromised or immune-reconstituted female BALB-c nude mice. Each subject was treated with a single intratumoral administration of 100 µL PBS formulation vehicle or 100X virus stock of each applicable recombinant virus to result in a virus mass dose of 5 × 10^6^ plaque-forming units (pfu) per animal when tumors reached approximately 200 mm^3^. [^18^F]-FDG PET/CT images (200 µCi/subject) were acquired 7–8 weeks post tumor cell inoculation. Select representative PET/CT images are as follows: immunocompromised vehicle control (**A**), immune-reconstituted vehicle control (**B**), TPV/eGFP immunocompromised (**C**), and TPV/eGFP immune-reconstituted (**D**). Panel (**E**) (TPV/eGFP immunocompromised subject in sagittal, AP, and transverse planes from left to right) demonstrates the multicompartmental composition of tumor mass observed in many study subjects; white arrows indicate tumor location.

**Figure 7 biomedicines-12-01834-f007:**
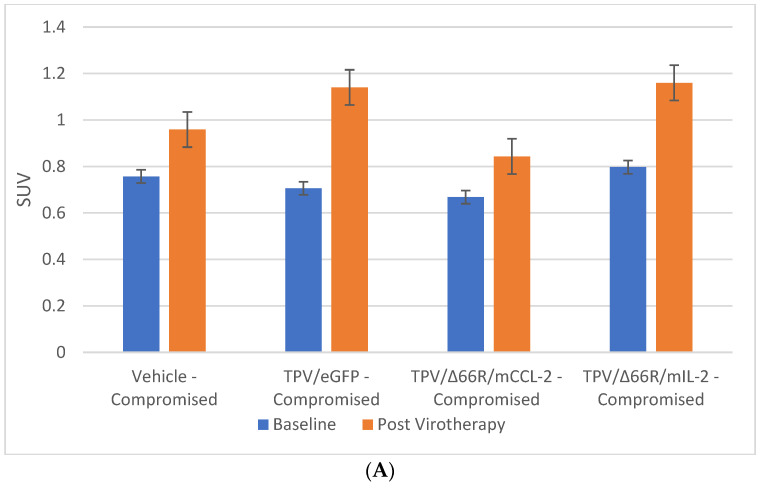
Immunocompromised group mean tumor [^18^F]-FLT PET/CT-derived standardized uptake value (SUV) and total lesion proliferation (TLP). BxPc-3 human PDAC cells were inoculated subcutaneously (SC) in the right flank region of immunocompetent female BALB-c nude mice. Each subject received a single inoculation of 5 × 10^6^ cells/subject. Each subject was treated with a single intratumoral administration of 100 µL PBS formulation vehicle or 100X virus stock of each applicable recombinant virus to result in a virus mass dose of 5 × 10^6^ plaque-forming units (pfu) per animal when tumors reached approximately 200 mm^3^. Baseline [^18^F]-FLT PET/CT images (200 µCi/subject) prior to virotherapy were acquired 3 weeks post BxPc-3 cell inoculation, and post-virotherapy images were collected at 7–8 weeks post inoculation. Bars show standard error of the mean (±1 SEM); SUV results for each respective group are shown in plot (**A**) and TLP results are show in panel (**B**). SUV and TLP inter- and intragroup statistical comparisons resulted in a lack of statistical significance for all comparisons except for immunocompromised vehicle-treated subject baseline to post-virotherapy results (*p* < 0.05, indicated by asterisk (*)).

**Figure 8 biomedicines-12-01834-f008:**
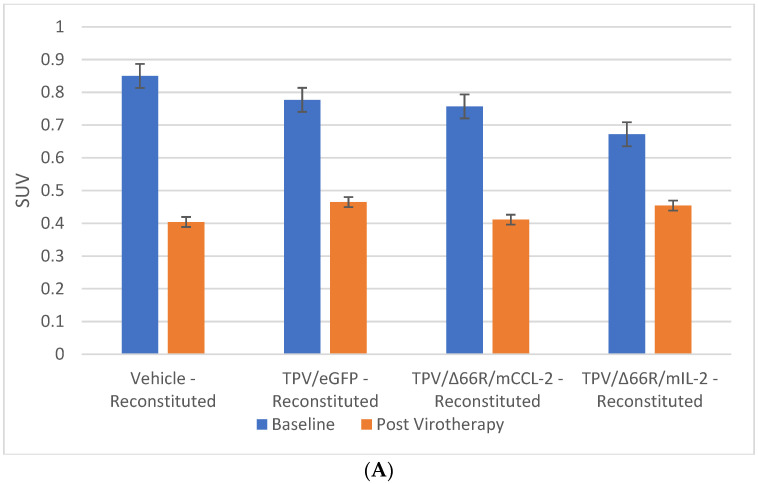
Immune-reconstituted group mean tumor [^18^F]-FLT PET/CT-derived standardized uptake value (SUV) and total lesion proliferation (TLP). BxPc-3 human PDAC cells were inoculated subcutaneously (SC) in the right flank region of immune reconstituted female BALB-c nude mice. Each subject received a single inoculation of 5 × 10^6^ cells/subject. Each subject was treated with a single intratumoral administration of 100 µL PBS formulation vehicle or 100X virus stock of each applicable recombinant virus to result in a virus mass dose of 5 × 10^6^ plaque-forming units (pfu) per animal when tumors reached approximately 200 mm^3^. Baseline [^18^F]-FLT PET/CT images (200 µCi/subject) prior to virotherapy were acquired 3 weeks post BxPc-3 cell inoculation, and post-virotherapy images were collected at 7–8 weeks post inoculation. Bars show standard error of the mean (±1 SEM); SUV results for each respective group are shown in plot (**A**) and TLP results are show in plot (**B**). SUV inter- and intragroup statistical comparisons resulted in a lack of statistical significance for all comparisons. TLP inter- and intragroup statistical comparisons resulted in a lack of statistical significance for all comparisons within immune reconstituted subjects; TLP results for post-virotherapy vehicle and TPV/∆66R/mCCL-2-treated subjects were statistically different (*p* < 0.05, indicated by asterisk (*)) from immunocompromised vehicle-treated control subjects.

**Figure 9 biomedicines-12-01834-f009:**
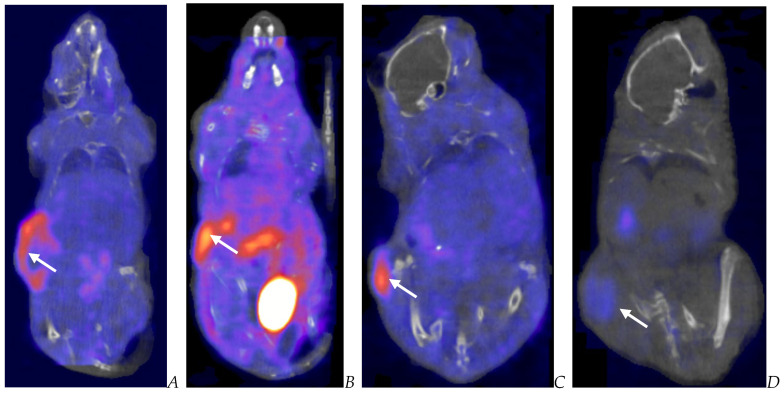
Post-virotherapy [^18^F]-FLT PET/CT images of BxPc-3 PDAC human tumor xenografts in immunocompromised and immune-reconstituted BALB-c nude mice. BxPc-3 human PDAC cells (5 × 10^6^ cells/subject) were inoculated subcutaneously (SC) in the right flank region of immunocompetent female BALB-c nude mice. Each subject was treated with a single intratumoral administration of 100 µL PBS formulation vehicle or 100X virus stock of each applicable recombinant virus to result in a virus mass dose of 5 × 10^6^ plaque-forming units (pfu) per animal when tumors reached approximately 200 mm^3^. [^18^F]-FLT PET/CT images (200 µCi/subject) were acquired 7–8 weeks post tumor cell inoculation. Select representative PET/CT images are as follows: immunocompromised vehicle control (**A**), immune-reconstituted vehicle control (**B**), TPV/eGFP immunocompromised (**C**), and TPV/eGFP immune reconstituted (**D**); white arrows indicate tumor location.

**Figure 10 biomedicines-12-01834-f010:**
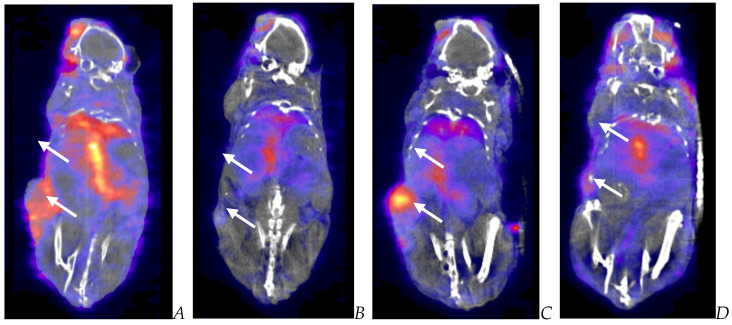
[^125^I]-anti-eGFP or [^125^I]-anti-mCherry antibody SPECT/CT images of BxPc-3 PDAC human tumor xenografts in immunocompromised and immune-reconstituted BALB-c nude mice. BxPc-3 human PDAC cells (5 × 10^6^ cells/subject) were inoculated subcutaneously (SC) in the right flank region of immunocompetent female BALB-c nude mice. Each subject was treated with a single intratumoral administration of 100 µL PBS formulation vehicle or 100X virus stock of each applicable recombinant virus to result in a virus mass dose of 5 × 10^6^ plaque-forming units (pfu) per animal when tumors reached approximately 200 mm^3^. SPECT/CT images were acquired during weeks 7–8, 48 h post imaging agent administration (200 µCi/subject). Select representative SPECT/CT images are as follows: immunocompromised vehicle control (A), immune reconstituted vehicle control (B), TPV/eGFP immunocompromised (C), and TPV/eGFP immune reconstituted (D); white arrows indicate tumor location.

**Figure 11 biomedicines-12-01834-f011:**
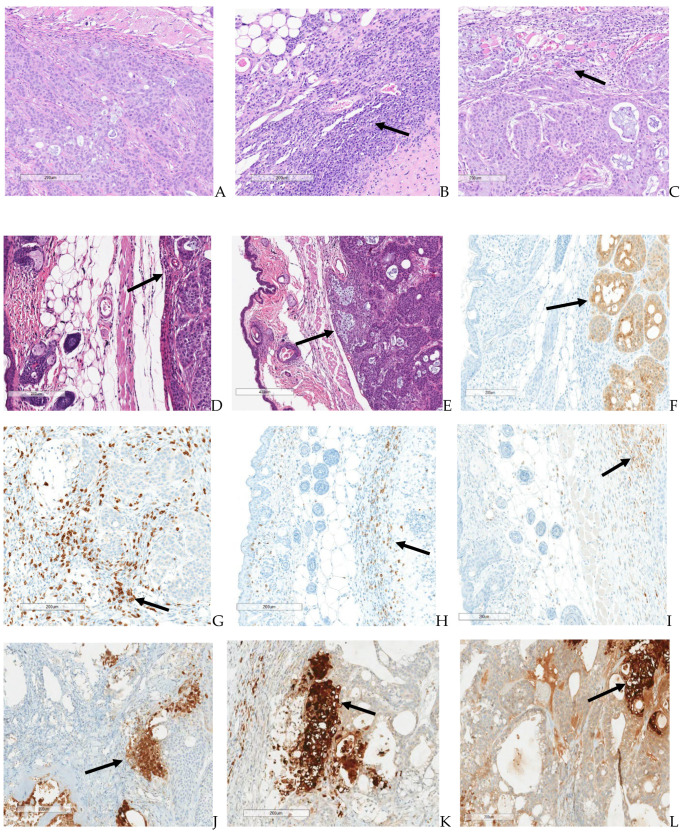
Histopathology and immunohistochemistry photomicrograph images of BxPc-3 PDAC human tumor xenografts in immunocompromised and immune-reconstituted BALB-c nude mice. Select representative photomicrograph images as follows: (**A**) H&E-stained, immunocompromised vehicle control with healthy adenocarcinoma; (**B**) H&E-stained, immune-reconstituted vehicle control adenocarcinoma with areas of necrosis and inflammatory cell infiltrates (black arrow); (**C**) H&E-stained, immunocompromised TPV/eGFP adenocarcinoma with areas of necrosis and inflammatory cell infiltrates (black arrow); (**D**) H&E-stained, immunocompromised TPV/∆66R/mIL-2 adenocarcinoma with areas of necrosis and inflammatory cell infiltrates (black arrow); (**E**) H&E-stained, immunocompromised TPV/∆66R/mCCL-2 adenocarcinoma with areas of necrosis and inflammatory cell infiltrates (black arrow); (**F**) caspase-stained, immunocompromised TPV/∆66R/mCCL-2 adenocarcinoma with areas of positive staining (black arrow); (**G**) CD-3-stained, immunocompromised TPV/∆66R/mCCL-2 adenocarcinoma with areas of positive staining (black arrow); (**H**) CD-4-stained, immunocompromised TPV/∆66R/mIL-2 adenocarcinoma with areas of positive staining (black arrow); (**I**) CD-68-stained, immunocompromised TPV/∆66R/mCCL-2 adenocarcinoma with areas of positive staining (black arrow); (**J**) GFP-stained, immunocompromised TPV/eGFP adenocarcinoma with areas of positive staining demonstrating viral replication and transgene expression (black arrow); (**K**) mCherry-stained, immunocompromised TPV/∆66R/mCCL-2 adenocarcinoma with areas of positive staining demonstrating viral replication and transgene expression (black arrow); and (**L**) mCherry-stained, immunocompromised TPV/∆66R/mIL-2 adenocarcinoma with areas of positive staining demonstrating viral replication and transgene expression. All bars = 200 µm, except for image E bar = 400 µm.

**Table 1 biomedicines-12-01834-t001:** [^18^F]-FDG PET/CT tumor region metabolic activity post-virotherapy assessment summary.

Immune System Status	Oncolytic Virotherapy	Mean Tumor Volume (mm^3^)	SUV	TLG
Immunocompromised	NA—control	428.62	0.165	0.048
Immunocompromised	TPV/eGFP	348.68	0.227	0.041
Immunocompromised	TPV/∆66R/m-IL-2/mCherry	300.44	0.281	0.029
Immunocompromised	TPV/∆66R/m-CCL-2/mCherry	235.50	0.241	0.045
Immune reconstituted	NA—control	39.99	0.328	0.195
Immune reconstituted	TPV/eGFP	117.62	0.231	0.033
Immune reconstituted	TPV/∆66R/m-IL-2/mCherry	145.87	0.333	0.033
Immune reconstituted	TPV/∆66R/m-CCL-2/mCherry	168.06	0.258	0.035

**Table 2 biomedicines-12-01834-t002:** [^18^F]-FLT PET/CT tumor region proliferation activity post-virotherapy assessment summary.

Immune System Status	Oncolytic Virotherapy	Mean Tumor Volume (mm^3^)	SUV	TLP
Immunocompromised	NA—control	428.62	0.959	0.404
Immunocompromised	TPV/eGFP	348.68	1.14	0.237
Immunocompromised	TPV/∆66R/m-IL-2/mCherry	300.44	1.16	0.139
Immunocompromised	TPV/∆66R/m-CCL-2/mCherry	235.50	0.843	0.151
Immune reconstituted	NA—control	39.99	0.404	0.017
Immune reconstituted	TPV/eGFP	117.62	0.465	0.745
Immune reconstituted	TPV/∆66R/m-IL-2/mCherry	145.87	0.454	0.051
Immune reconstituted	TPV/∆66R/m-CCL-2/mCherry	168.06	0.411	0.049

**Table 3 biomedicines-12-01834-t003:** SPECT/CT tumor region TPV transgene expression post-virotherapy assessment summary.

Immune System Status	Oncolytic Virotherapy	Mean Tumor Volume (mm^3^)	Percent Injected Dose/Gram Tissue
Immunocompromised	NA—control	428.62	1.95
Immunocompromised	TPV/eGFP	348.68	3.14
Immunocompromised	TPV/∆66R/m-IL-2/mCherry	300.44	2.49
Immunocompromised	TPV/∆66R/m-CCL-2/mCherry	235.50	2.09
Immune reconstituted	NA—control	39.99	1.22
Immune reconstituted	TPV/eGFP	117.62	1.58
Immune reconstituted	TPV/∆66R/m-IL-2/mCherry	145.87	1.69
Immune reconstituted	TPV/∆66R/m-CCL-2/mCherry	168.06	1.41

**Table 4 biomedicines-12-01834-t004:** [^125I^]-anti-eGFP and [^125I^]-anti-mCherry antibody QWBA tumor region TPV transgene expression post-virotherapy assessment summary.

Immune System Status	Oncolytic Virotherapy	Mean Tumor Volume (mm^3^)	Nanogram Antibody/Gram Myocardium	Nanogram Antibody/Gram Tumor
Immunocompromised	NA—control	428.62	110	441
Immunocompromised	TPV/eGFP	348.68	158	528
Immunocompromised	TPV/∆66R/m-IL-2/mCherry	300.44	38	156
Immunocompromised	TPV/∆66R/m-CCL-2/mCherry	235.50	36	138
Immune reconstituted	NA—control	39.99	88	325
Immune reconstituted	TPV/eGFP	117.62	82	281
Immune reconstituted	TPV/∆66R/m-IL-2/mCherry	145.87	50	149
Immune reconstituted	TPV/∆66R/m-CCL-2/mCherry	168.06	48	81

**Table 5 biomedicines-12-01834-t005:** Immunocompromised vehicle, TPV/eGFP, TPV/∆66R/mCCL-2, and TPV/∆66R/mIL-2 treatment histopathological summary.

Treatment	Vehicle	TPV/eGFP	TPV/∆66R/mCCL-2	TPV/∆66R/mIL-2
No. Animals per Group	6	6	6	6
**Tumor transplant** **(number examined)**	(6)	(6)	(6)	(6)
Adenocarcinoma; malignant, primary				
Present	6	6	6	6
Caspase positive				
Moderate	-	-	1	-
Marked	6	6	5	6
CD3 positive				
Minimal	2	2	-	2
Mild	4	4	1	3
Moderate	-	-	4	1
Marked	-	-	1	-
CD4 positive				
Minimal	4	5	-	5
Mild	2	-	4	1
Moderate	-	-	2	-
CD68 positive				
Minimal	-	-	-	1
Mild	4	5	1	4
Moderate	2	1	5	1
CD8 positive				
Minimal	2	3	1	1

**Table 6 biomedicines-12-01834-t006:** Immune-reconstituted vehicle, TPV/eGFP, TPV/∆66R/mCCL-2, and TPV/∆66R/mIL-2 treatment histopathological summary.

Treatment	Vehicle	TPV/eGFP	TPV/∆66R/mCCL-2	TPV/∆66R/mIL-2
No. Animals per Group	6	6	6	6
**Tumor transplant** **(number examined)**	(5)	(6)	(6)	(6)
Adenocarcinoma; malignant, primary				
Present	-	2	-	1
Caspase positive				
Minimal	2	2	-	-
Moderate	-	1	-	-
Marked	-	1	-	1
CD3 positive				
Mild	1	1	-	-
Moderate	3	3	1	2
Marked	1	2	5	4
CD4 positive				
Minimal	-	1	-	-
Mild	-	2	1	4
Moderate	4	3	4	2
Marked	1	-	1	-
CD68 positive				
Minimal	-	-	1	-
Mild	-	-	1	1
Moderate	-	5	4	4
Marked	5	1	-	-
CD8 positive				
Minimal	1	-	-	1

## Data Availability

Data are contained within the article and Appendix A.

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
