# Peer review of "Oncolytic Tanapoxvirus Variants Expressing mIL-2 and mCCL-2 Regress Human Pancreatic Cancer Xenografts in Nude Mice†"

_biomedicines, 2024, doi:10.3390/biomedicines12081834_

Round 1

Reviewer 1 Report

Comments and Suggestions for Authors

In the reviewed manuscript the authors showed that  tanapoxvirus  recombinants TPV/∆66R/mIL-2 and TPV/∆66R/mCCL-2 were more effective in regression of BxPc-3 pancreatic tumor volume and tumor proliferation rate as compared to TPV/eGFP control recombinant. According to above the authors postulated that TPV recombinants expressing mCCL2 and mIL-2 demonstrated a therapeutic effect via regression of BxPc-3 tumor  xenografts and TPV has oncolytic potency against pancreatic ductal adenocarcinoma.

The research is interesting but lacks statistical analysis on most of the charts/figures. Therefore, it is unclear whether the observed changes are actually biologically significant.

The construction of TPV recombinants should be verify by means of appropriate tests.  For example TPV/∆66R/m-IL-2/mCherry should be presented using also in vitro model and confocal microcopy techniques to confirm the effectiveness of transduction.

All figures should be more carefully prepared with appropriate graphics programs. Currently, they are of poor quality and rather unattractive.

Histopathology and immunohistochemistry  should be verify by immunofluorescence for co-localization tanapoxvirus  recombinants and selected markers of apoptosis and/or necroptosis and/or pyroptosis.

Please note that the tanapoxvirus  recombinants can induced different PCD pathways and the authors should resolve which type of death is induced

Reviewer 2 Report

Comments and Suggestions for Authors

The manuscript shows an interesting topic regarding the use of Oncolytic Tanapoxvirus as a PDAC therapeutic in xenograft mouse models. It shows a potential strategy for PDAC.

However, the entire result section should be rewritten with well-arranged figures. The current versions are totally hard to follow.

Major Comments:

For all result part.

It should have 1-2 sentences in each part to introduce what the author wants to do and also the method used in each.

The figure should be rearranged based on the content of each part for the reader to follow. Not arranged by the methods.

Figure 1. It should have a mark to indicate which lines were indicated.

The unit should be fixed: not write as mm^3

Study day -> Time (day)

Figure 4,5. What is SUV unit?

The arrow in the figure with the CT image( for example, Figure 6, 9,11) was shown in the wrong area.

There are so many figures in the main text. The author Can rearrange and move in sufficient figures to supplement data.

The table can be moved to supplement the data.

Minor comment

The title is too long and should be rephrased.

Page 2, line 74. ’Epigenetic modifications have 74 also been associated with PDAC”: so general. Should add more information.

Result

Part 1. Replication kinetics of TPV in human pancreatic cancer cells: The figure is not mentioned in the text.

Page 25-26, lines 639-693. Missing indicates the figure in the text.

Comments on the Quality of English Language

Must be improved

Author Response

Journal- Biomedicines

Biomedicines-3003347

Title- Oncolytic Tanapoxvirus Variants Expressing mIL-2 and mCCL-2 Regress Human Pancreatic Cancer Xenografts in Immunocompromised and Immune Reconstituted CAnN.Cg-Foxn1nu/Crl Nude Mice

Authors- Dr. Scott Haller & Dr. Karim Essani

Regarding- Author(s) Response to Reviewer 2, Review Report (Round 1)

Date- June 03, 2024

Comments and Suggestions for the Authors:

Reviewer 2, Major Comment Item 1-

For all result part.

It should have 1-2 sentences in each part to introduce what the author wants to do and also the method used in each.

Author(s) Response, Major Comment Item 1-

The Results section of scientific manuscripts intentionally do not include abbreviated or elaborative reiteration of details related experimental methods to avoid unnecessary redundancy of text in manuscripts as the applicable details related to experimental methods are described in the Materials and Methods Section of articles as standard practice.

This standard format/presentation of Material and Methods, as well as Results in our submitted article follow this format and we, respectfully, submit this as response to Reviewer 2, Item 1.

Reviewer 2, Major Comment Item 2-

The figure should be rearranged based on the content of each part for the reader to follow. Not arranged by the methods.

Author(s) Response, Major Comment Item 2-

The figures in our article are not arranged by the method used to generate each data set and the applicable results.  The figures in our article are presented a logical manner starting with in vitro assay results, followed by in vivo assay results and conclude with ex vivo assay results.  It should also be noted that the order in which are results are presented also follow the logical cadence in which our studies were conducted temporally as we moved through each phase of our overall study program; being in vitro assays to inform appropriate design and execution of our in vivo assays, followed by ex vivo assays which require tissue samples generated through conduct of the in vivo phase of the study.

We respectfully disagree with a proposed change to the arrangement of or order in which our results are presented in our article.

Reviewer 2, Major Comment Item 3-

Figure 1. It should have a mark to indicate which lines were indicated.

The unit should be fixed: not write as mm^3

Study day -> Time (day)

Author(s) Response, Major Comment Item 3-

Figure 1 is presented in manner that is consistent with all other figures in our article which present tumor volume.  The tumor volume curve/line for each applicable control group in each subset of study subjects, subsets being differentiated by immune state as immunocompromised or immune reconstituted, is described in the figure legend for all figures based on the color of the line and shape of the symbol for each datapoint corresponding to the control group which allows the reader to differentiate the control vs. treatment groups in each panel presented in the figures. It should also be noted that the color and shape of the of the line and datapoint symbols are consistent within each figure.  This allows the reader to be informed as to the origin of each dataset while reducing overall length of the figure legends.  

The unit for reporting tumor volume as “mm^3” is a universally accepted presentation for the unit of millimeters cubed to report volumetric data. 

The reported unit of Study Day on the X-axis defines the in vivo temporal relationship from start of study, tumor cell inoculation, for each group of animals to completion of study, euthanasia and collection of required tissues.   

We feel the presentation of the data within the figures, axis naming convention, reporting units and descriptive text in each figure legend effectively, yet succinctly provide the reader with sufficient detail to understand and interpret the results presented in all figures throughout the article. 

Reviewer 2, Major Comment Item 4-

Figure 4,5. What is SUV unit?

Author(s) Response, Major Comment Item 4-

The unit of SUV is a standard unit for reporting data generated via PET or other molecular imaging platforms.  The details for reporting as SUV were described our original submission Methods and Materials Section “PET/CT Imaging” of our article.  Relevant text is copied here for completeness.  “Fixed volume regions-of-interest (ROIs) were used to quantify total radioactivity in tumor and heart for each animal at each time point; results were reported as Standardized Uptake Value (SUV ((decay-corrected activity of tissue volume)/(injected activity/body mass)) for each ROI.  ROIs for primary analysis included tumor and heart; heart was used as surrogate to determine systemic activity; heart was used as surrogate to determine systemic activity.”

Reviewer 2, Major Comment Item 5-

The arrow in the figure with the CT image ( for example, Figure 6, 9,11) was shown in the wrong area.

Author(s) Response, Major Comment Item 5-

Initial reformatting of the full article through Biomedicines resulted in shift of demarcations (arrows) of relevant region of interest or specific tissue area being discussed in results text and figure legends for all PET, SPECT, QWBA and histopathology images; this has been corrected, although the relocation of arrows to the intended/proper location in all figures does not display as a tracked change.

Reviewer 2, Major Comment Item 6-

There are so many figures in the main text. The author Can rearrange and move in sufficient figures to supplement data.

The table can be moved to supplement the data.

Author(s) Response, Major Comment Item 6-

While we feel the figures in our article provide relevant information and graphical and visual presentation of our data, we agree there is potential to remove select figures from the main article and present as Supplemental data.

Our suggestion for potential relocation of figures to Supplemental is as follows: Figure 10, Figure 12, Figure 13, Table 7 and Figure 15.

We can proceed with removal of these figures and table, revise main article text as applicable to refer to supplemental data and submit as a separate file for supplemental data, if these changes satisfy Reviewer 2’s comment in this regard.  Our intent would be complete these revisions following response from second round of review and prior to submitting final article for publication.

Reviewer 2, Minor Comment Item 1-

The title is too long and should be rephrased.

Author(s) Response, Minor Comment Item 1-

We respectfully disagree with revision/rephrasing of the title given the current titles contains necessary detail to appropriately relay the full scope of our study and outcome.

Reviewer 2, Minor Comment Item 2-

Page 2, line 74. ’Epigenetic modifications have 74 also been associated with PDAC”: so general. Should add more information.

Author(s) Response, Minor Comment Item 2-

The inclusion of “74” in Reviewer 2 comment in this item does not accurately reflect the actual text in the article.  Reviewer 2 appears to be including the line number as part of the text, which is not intended, nor is it commented on for any other line in the article which is displayed in the same manner throughout the paper.

The actual text for this sentence reads as intended by the authors as “Epigenic modifications have also been associated with PDAC[19-29].”

Reviewer 2, Minor Comment Item 3-

Result

Part 1. Replication kinetics of TPV in human pancreatic cancer cells: The figure is not mentioned in the text.

Author(s) Response, Minor Comment Item 3-

There is no mention of a figure or intent by us as authors to include a figure in this results section.  Descriptive text to describe our results of in vitro assays assessing TPV replication kinetics is our intended presentation of our results.

Reviewer 2, Minor Comment Item 4-

Page 25-26, lines 639-693. Missing indicates the figure in the text.

Author(s) Response, Minor Comment Item 4-

We are unable to determine the intended context from Reviewer 2 for this item.  The referred to section of the article between lines 639 – 693 is primarily comprised of Table 6 containing results of immune reconstituted study groups and Figure 14 which displays various histopathology and IHC images and figure legend text. 

Comments on the Quality of the English Language, Review 2-

Must be improved.

Author(s) Response

As author of this paper and native English, speaking individuals since birth, or since permanent relocation to the US for more than 40 years, we find this comment somewhat befuddling.  Following careful and unbiased review of our article for consistent and appropriate use of written English, we find no issues with English throughout the article.  We respectfully disagree. 

Round 2

Reviewer 2 Report

Comments and Suggestions for Authors

The manuscript has undergone fewer improvements than the first version. However, the entire manuscript needs rewriting, especially the results section and figures, which need to be rearranged. The current version is difficult to follow.

major comment:

  • The results section requires 1-2 introductory sentences for each part to outline the objectives and methods used.
  • Subtitles in each section should reflect the results obtained, not the methods used.
  • Figures should be reorganized according to the content of each section rather than by the methods used to enhance readability.
  • Lines 200-208 should provide supporting data for this result.
  • In Figure 1, markers should indicate which lines are referenced.
  • Units should be standardized (e.g., use "mm^3" consistently).
  • Replace "Study day" with "Time (day)."
  • Consider moving excessive figures to supplementary data to streamline the main text.
  • Tables can also be moved to supplement the data.

minor comments

  • The title is overly long and should be rephrased.
  • Introduction sections typically do not have subtitles in research papers; consider removing them.
  • Lines 61-116 contain extraneous information not directly relevant to this study; these should be reorganized or removed.

Comments on the Quality of English Language

should be rearranged and rewrite

Author Response

Journal- Biomedicines

Biomedicines-3003347

Title- Oncolytic Tanapoxvirus Variants Expressing mIL-2 and mCCL-2 Regress Human Pancreatic Cancer Xenografts in Nude Mice

Authors- Dr. Scott Haller & Dr. Karim Essani

Regarding- Author(s) Response to Reviewer 2, Review Report (Round 2)

Date- July 01, 2024

Thank you sincerely for taking the time to this review manuscript.  Please find detailed responses to each Round 2 comment and suggestion below.

Comments and Suggestions for the Authors:

Reviewer 2, Major Comment Item 1-

The results section requires 1-2 introductory sentences for each part to outline the objectives and methods used.

Author(s) Response, Major Comment Item 1-

Where applicable, short introductory sentences were added or previous sentence structure revised to results sections to clearly state the objective of individual assays/assessments.  Track changes indicate all such additions.

Reviewer 2, Major Comment Item 2-

Subtitles in each section should reflect the results obtained, not the methods used.

Author(s) Response, Major Comment Item 2-

We have carefully reviewed the subtitle structure used in our article and compared against contemporaneous articles published in the Biomedicines Journal.  While we respect the perspective of Reviewer 2, we find the structure of subtitles is consistent recently published articles in Biomedicines and also consistent with standard subtitle structure used across scientific publications in many scientific journals.  See the following recent Biomedicines publications as examples of subtitle structure and consistency with the structure used in our article: https://www.mdpi.com/2227-9059/9/3/247, https://www.mdpi.com/2227-9059/9/11/1714.

Reviewer 2, Major Comment Item 3-

Figures should be reorganized according to the content of each section rather than by the methods used to enhance readability.

Author(s) Response, Major Comment Item 3-

While we respect the perspective of Reviewer 2, the figures in our article are not arranged by the method used to generate each data set and the applicable results.  The figures in our article are presented a logical manner starting with in vitro assay results, followed by in vivo assay results and conclude with ex vivo assay results.  It should also be noted that the order in which our  results are presented follow the logical cadence in which our studies were conducted temporally as we moved through each phase of our overall study program; being in vitro assay results used to inform appropriate in vivo study design, execution of our in vivo assays, followed by ex vivo assays which require ex vivo tissue samples generated through conduct of the in vivo phase of the study.

We respectfully disagree with a proposed change to the arrangement of the order in which our results are presented in our article.

Reviewer 2, Major Comment Item 4-

Lines 200-208 should provide supporting data for this result.

Author(s) Response, Major Comment Item 4-

While we respect the perspective of Reviewer 2, the study results presented in the text contained in lines 200 – 208 are supported with data and appropriately referenced to the applicable Figures in the “Treatment of BxPc-3 xenografts with Tanapoxvirus recombinants in vivo” results section of our article.   

Reviewer 2, Major Comment Item 5-

Figure 1. It should have a mark to indicate which lines were indicated.

Author(s) Response, Major Comment Item 5-

While we respect the perspective of Reviewer 2, and as we had responded to Round 1 comments, after a thorough, secondary review, we find Figure 1 is presented in manner that is consistent with all other figures in our article which present tumor volume.  The tumor volume curve/line for each applicable control group in each subset of study subjects, subsets being differentiated by immune state as immunocompromised or immune reconstituted, is described in the figure legend for all figures based on the color of the line and shape of the symbol for each datapoint corresponding to the control group which allows the reader to differentiate the control vs. treatment groups in each panel presented in the figures. It should also be noted that the color and shape of the of the line and datapoint symbols are consistent within each figure.  This allows the reader to be informed as to the origin of each dataset while reducing overall length of the figure legends.

Reviewer 2, Major Comment Item 6-

Units should be standardized (e.g., use "mm^3" consistently).

Author(s) Response, Major Comment Item 6-

We agree and appreciate Reviewer 2 commenting regarding reporting units.  We have revised the Y-axis reporting unit to read as “Tumor Volume (mm3)” for Figures 1, 2, and 3.  This reporting unit description is consistent with all other plots or tables displaying volumetric data in the article to avoid any potential confusion regarding reporting unit being cubic millimeters with superscript “3”. 

Please note, given the plots are embedded in the body of the article, revisions to the axis titles do not show as track changes.

Reviewer 2, Major Comment Item 7-

Replace "Study day" with "Time (day)."

Author(s) Response, Major Comment Item 7-

We agree and appreciate Reviewer 2 commenting regarding potential revision to X-axis title for all plots displaying time relative to study.  All applicable plots have been revised to have X-axis title read as “Time (Day)”. 

Reviewer 2, Major Comment Item 8 & 9-

Consider moving excessive figures to supplementary data to streamline the main text.

Tables can also be moved to supplement the data.

Author(s) Response, Major Comment Item 4-

We agree and appreciate Reviewer 2 commenting regarding potentially moving figures and/or tables to supplemental data. 

We have moved the following Figures and Tables to Supplemental:  Figure 10, Figure 12, Figure 13, Table 7 and Figure 15.  We have uploaded a separate file containing each Figure, with applicable figure legend, and Table.  

Reviewer 2, Minor Comment Item 1-

The title is too long and should be rephrased.

Author(s) Response, Minor Comment Item 1-

We have revised the main and short titles of our article to a more succinct title to address this minor comment from Reviewer 2.  Revised titles can be found in the revised version of our article and as the title of our article as written in this response document to Round 2 comments.

Reviewer 2, Minor Comment Item 2 & 3-

Introduction sections typically do not have subtitles in research papers; consider removing them.

Lines 61-116 contain extraneous information not directly relevant to this study; these should be reorganized or removed.

Author(s) Response, Minor Comment Item 2-

We appreciate Reviewer 2 commenting regarding consideration of removing subtitles in the Introduction Section of our article and reviewing lines 61-116 for potential consolidation.  

We have removed subtitles throughout the introduction section and also revised text contained in lines 61-116 to present relevant background to support our research objectives and rationale for study design while decreasing total word count in the article.